# Mechanical heterogeneity along single cell-cell junctions is driven by lateral clustering of cadherins during vertebrate axis elongation

Robert J Huebner[1†], Abdul Naseer Malmi-Kakkada[2,3†], Sena Sarıkaya[1], Shinuo Weng[1], D Thirumalai[2]*, John B Wallingford[1]*

[1]Department of Molecular Biosciences, University of Texas, Austin, United States; [2]Department of Chemistry, University of Texas, Austin, United States; [3]Department of Chemistry and Physics, Augusta University, Augusta, Georgia

**Abstract** Morphogenesis is governed by the interplay of molecular signals and mechanical forces across multiple length scales. The last decade has seen tremendous advances in our understanding of the dynamics of protein localization and turnover at subcellular length scales, and at the other end of the spectrum, of mechanics at tissue-level length scales. Integrating the two remains a challenge, however, because we lack a detailed understanding of the subcellular patterns of mechanical properties of cells within tissues. Here, in the context of the elongating body axis of *Xenopus* embryos, we combine tools from cell biology and physics to demonstrate that individual cell-cell junctions display finely-patterned local mechanical heterogeneity along their length. We show that such local mechanical patterning is essential for the cell movements of convergent extension and is imparted by locally patterned clustering of a classical cadherin. Finally, the patterning of cadherins and thus local mechanics along cell-cell junctions are controlled by Planar Cell Polarity signaling, a key genetic module for CE that is mutated in diverse human birth defects.

*For correspondence:
dave.thirumalai@gmail.com (DT);
wallingford@austin.utexas.edu (JBW)

†These authors contributed equally to this work

Competing interests: The authors declare that no competing interests exist.

## Introduction

The establishment and maintenance of animal form involves the control of physical forces by molecular systems encoded in the genome, and the elongation of an animal's head-to-tail body axis is a long-studied paradigm for understanding morphogenesis (*Guillot and Lecuit, 2013*). This essential step in the construction of a new embryo is driven by an array of morphogenetic engines, including an evolutionarily ancient suite of collective cell behaviors termed convergent extension (*Figure 1A*; *Figure 1—figure supplement 1*; *Huebner and Wallingford, 2018*). Critically, failure of axis elongation does not simply result in a shorter embryo, but rather has catastrophic consequences, and defects in convergent extension in mammals, including humans, results in lethal birth defects (*Wallingford et al., 2013*).

The biomechanics of convergent extension (CE) and axis elongation more generally have been studied across diverse length scales, providing several key insights (*Davidson, 2017*; *Mongera et al., 2019*; *Stooke-Vaughan and Campàs, 2018*). At the tissue scale, these include quantitative descriptions of patterned macroscopic stiffening (*Moore et al., 1995*; *Zhou et al., 2009*), tissue-scale jamming transitions (*Mongera et al., 2018*; *Serwane et al., 2017*), and fluid-like multicellular flows (*Bénazéraf et al., 2010*; *Lawton et al., 2013*). At smaller length scales, laser ablation studies have provided insights into the patterns of junctional tension within these tissues (*Fernandez-Gonzalez et al., 2009*; *Rauzi et al., 2008*; *Shindo and Wallingford, 2014*). Finally, a more granular examination of the mechanics of CE has been made possible by the use of theoretical

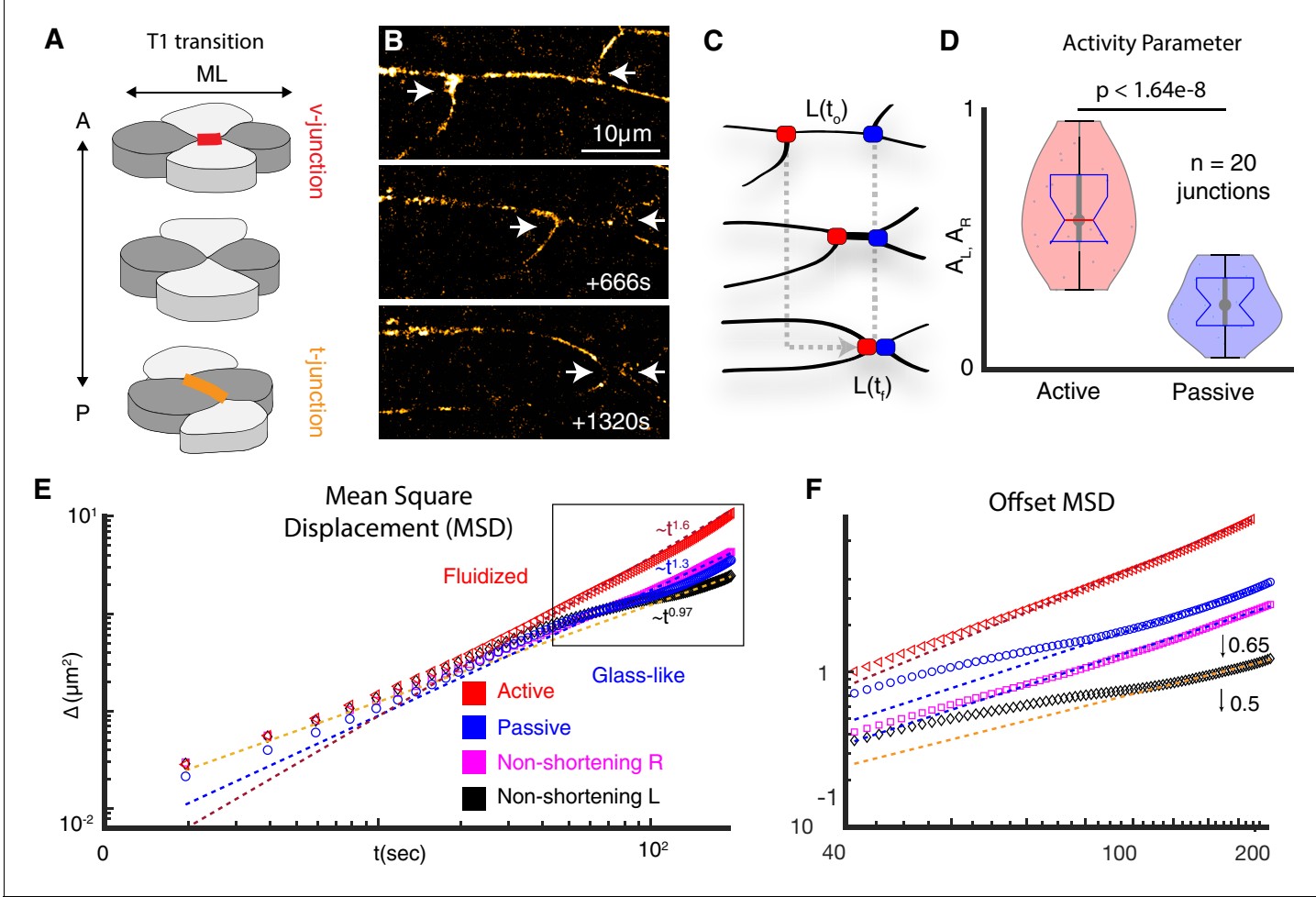

**Figure 1.** Vertices bounding shortening v-junctions are physically asymmetric and display heterogeneous fluid and glass-like dynamics. (**A**) A four cell T1 transition with mediolaterally (ML)-aligned 'v-junctions' (red) and anterior-posterior (A/P) aligned t-junctions (orange) indicated. (**B**) Frames from time-lapse showing vertex movements of a v-junction; arrows highlight vertices. Frames were acquired at a z-depth of 5 µm above the ECM/coverslip and with a time interval of 2 s. (**C**) Schematic of asymmetric vertex movements from B; active = red; passive = blue. (**D**) Vertex motion quantified by the activity parameter, as described in Appendix, Section 1. (N = 42 vertices from 20 embryos; t-test p value is shown). (**E**) MSD reveals active vertices' persistent superdiffusive movement (red); passive vertices exhibit intermediate time slowdown (blue). Pink and black display MSD for left and right non-shortening junctions. MSD is described in Appendix, Section 2. (**F**) MSD from boxed region in E is shown with traces offset for clarity (0.5 for left; 0.65 for right)(N = 20 vertices from 10 embryos).

The online version of this article includes the following figure supplement(s) for figure 1:

**Figure supplement 1.** Schematics of *Xenopus* development.

**Figure supplement 2.** Mean squared displacement measured from multiple frames of reference.

**Figure supplement 3.** Extended analysis of vertex glass-like dynamics.

modeling (*Alt et al., 2017*; *Fletcher et al., 2017*; *Merkel and Manning, 2017*), with recent innovations continuing to improve these models (e.g. *Staddon et al., 2019*).

However, our still-emerging understanding of the cell biology of CE continues to present new challenges to understanding its biomechanics. For example, CE is driven by a combination of lamellipodia-based cell crawling by laterally positioned cells and by junction contraction in medially positioned cells (*Sun et al., 2017*; *Williams et al., 2014*; *Huebner and Wallingford, 2018*). When considering the biomechanics however, very few models consider cell crawling (e.g. *Belmonte et al., 2016*), with the majority of models addressing only junction contraction. In addition, these junction-contraction models consistently consider individual cell-cell junctions to be mechanically homogenous along their length (*Alt et al., 2017*; *Fletcher et al., 2017*; *Merkel and Manning, 2017*), yet recent work in single cells suggests this approach may be limited. Indeed, there

is accumulating evidence that single cells' membranes can be mechanically heterogeneous (*Lieber et al., 2015*; *Shi et al., 2018*; *Strale et al., 2015*). Thus, the role of mechanical heterogeneity along individual cell-cell junctions during collective cell movement in vivo remains poorly defined.

In addition, we know comparatively little about the interplay of *subcellular* mechanical properties in vivo and the molecules that govern them. Resolving this disconnect is crucial, because CE in diverse systems is known to require complex spatial and temporal patterns of protein localization and dynamics along individual cell-cell junctions during morphogenesis. For example, the localization and turnover of actomyosin and cadherin adhesion proteins have been extensively quantified during *Drosophila* CE (*Blankenship et al., 2006*; *Fernandez-Gonzalez et al., 2009*; *Levayer and Lecuit, 2013*; *Rauzi et al., 2008*), as have similar patterns for the Planar Cell Polarity (PCP) proteins and actomyosin during vertebrate CE (*Butler and Wallingford, 2018*; *Kim and Davidson, 2011*; *Shindo and Wallingford, 2014*). However, the significance of these molecular patterns remains unclear because we lack a similarly granular understanding of subcellular mechanical properties and their dynamics, which ultimately explain the cell behaviors that drive vertebrate CE.

Here, we combine high-speed super-resolution microscopy with concepts rooted in soft matter physics to demonstrate that individual cell-cell junctions in the elongating vertebrate body axis display finely-patterned local mechanical heterogeneity along their length. To explore this unexpected finding, we developed a new theory for junction remodeling in silico and new tools for assessment of very local mechanics in vivo. Combining these, we show that sub-cellular mechanical heterogeneity is essential for CE and is imparted by cadherins via locally patterned intracellular (*cis-*) interactions. Finally, the local patterns of both cadherin clustering and heterogeneous junction mechanics are controlled by PCP signaling, a key regulatory module for CE that is mutated in diverse human birth defects.

## Results

### The dynamics of tricellular junction motion during CE suggest an unexpected mechanical heterogeneity at subcellular length scales

The elongating body axis of *Xenopus* embryos is a long-standing and powerful model system for studying PCP-dependent vertebrate CE (*Figure 1—figure supplement 1*; *Huebner and Wallingford, 2018*). *Xenopus* CE can be considered most simply in terms of four-cell neighbor exchanges in which mediolaterally aligned cell-cell junctions ('v-junctions') shorten, followed by the elongation of new, perpendicularly aligned junctions ('t-junctions')(*Figure 1A*). To gain deeper insights into this process, we used high-speed super-resolution imaging to establish a quantitative physical description of the motion of tricellular vertices bounding v-junctions (*Figure 1B*).

First, we found that v-junction shortening was dominated by the movement of a single 'active' vertex, while the other 'passive' vertex moved comparatively less (*Figure 1C,D*), similar to the asymmetry observed previously during CE in *Drosophila* epithelial cells (*Vanderleest et al., 2018*). To quantify the asymmetric movement of vertices in *Xenopus* mesenchymal cells, we used a fixed coordinate system and defined an activity parameter, A, as the ratio of the net distance moved by the vertices to the initial junction length, (*Figure 1C*; *Figure 1—figure supplement 2A*) (Appendix, Section 1). This analysis demonstrated significant differences in the motion of active and passive vertices (*Figure 1D*).

We then explored the physical basis of asymmetric motion in active and passive vertices using mean squared displacement (MSD)(SI Section 2). Our analysis revealed that active vertices consistently displayed a highly fluidized movement (i.e. super-diffusive)(*Figure 1E,F*, red). By contrast, passive vertices displayed the hallmarks of more-constrained, glass-like motion (i.e. defined by sub-diffusive movement with an intermediate time slowdown, as observed in colloidal systems); (*Kegel and van Blaaderen, 2000*; *Weeks et al., 2000*; *Figure 1E,F*, blue). The juxtaposition of liquid- and glass-like motion along a single cell-cell junction was interesting, because while fluid-to-glass phase transitions are known features at tissue-level length scales (*Angelini et al., 2011*; *Bi et al., 2015*; *Malmi-Kakkada et al., 2018*; *Sinha et al., 2020*), such transitions have not been reported at sub-cellular length scales during morphogenesis.

Given these surprising results, we also controlled for the possibility that image drift may interfere with our quantification. To this end, we quantified the motion of active and passive vertices using

two relative reference frames with respect to slowly moving landmarks within the tissue (*Figure 1—figure supplement 2D–F*). All three quantification strategies demonstrate that the asymmetry we observed was *not* a point-of-reference artifact (*Figure 1—figure supplement 2D–G*).

Finally, we confirmed the distinct patterns of motion in active and passive vertices using four additional physical metrics, the Van Hove function, the velocity auto-correlation function, the self-overlap parameter, and the fourth order susceptibility, $\chi_4(t)$ (SI sections 2-3). All four orthogonal approaches confirmed our finding that the active and passive vertices bounding individual v-junctions exhibit asymmetric dynamic behaviors, with one vertex displaying a fluid-like motion and the other, glass-like (*Figure 1—figure supplement 3*). Critically, this asymmetric behavior was specific to shortening dynamics of v-junctions, as the two vertices bounding non-shortening junctions in the same tissue were consistently symmetrical, both resembling passive vertices (*Figure 1E,F*, pink, black; *Figure 1—figure supplement 3*).

This physical analysis provided three important insights: First, glass-like dynamics previously observed only at tissue-length scales in morphogenesis also exist at the subcellular length scale of individual junctions. Second, the frequently invoked assumption of mechanical homogeneity along single cell-cell junctions, which underlies a wide swath of the biophysical work on morphogenesis, may not be valid. And finally, because only shortening junctions exhibited local mechanical heterogeneities, this phenomenon may be a specific and essential feature of convergent extension.

## A new physical model of cell-cell junction remodeling predicts asymmetric, local patterning of junction stiffness as an essential feature of convergent extension

The possibility of mechanical heterogeneity along single cell-cell junctions has important implications, as many biophysical approaches and in silico tools for understanding morphogenesis (e.g. laser cutting, vertex models) assume that junctions are mechanically homogeneous along their length. We therefore developed a new theoretical framework for junction shortening that accommodates the possibility of local mechanical heterogeneity by independently modeling the movement of each vertex (*Figure 2A,B*; Appendix, Section 4–8).

Our model involves (i) a local junction stiffness (or elasticity) modeled using a spring element, which is consistent with the pulsatile relaxation of v-junctions observed in *Xenopus* CE (*Shindo and Wallingford, 2014*) (ii) a dynamic rest length, recently shown to be important for modeling CE (*Shindo et al., 2019*) (iii) a viscoelastic parameter, $k/\gamma$, dictated by the spring stiffness, $k$, and the friction at the vertices, $\gamma$; and (iv) a rest length exponent, $\psi$, which describes the time dependence of plastic displacement of the vertices modeled with a piston (*Figure 2A,B*; Appendix, Section 4-8).

Using this model, we explored parameter space to find variables in elastic and viscous deformation that can support effective shortening of the junction (Appendix, Sections 7, 8). As shown in the heatmap in *Figure 2C*, for a given rest length exponent, junctions failed to shorten if the viscoelastic parameter was equal and small for both the vertices (*Figure 2C*, red box; Appendix, Sections 7, 8). When the viscoelastic parameter was asymmetric, junctions shortened effectively (*Figure 2C*, gold box). Thus, at the level of binary outcome (i.e. shorten versus fail-to-shorten), our model suggests that CE requires mechanical heterogeneity along single v-junctions.

For a more stringent test, we compared the temporal dynamics of junction shortening in our model to those quantified in vivo from high-speed super-resolution movies (*Figure 2D,E*). In time-lapse data, the relaxation behavior of v-junctions collapsed into a self-similar pattern when normalized; relaxation became progressively more efficient over time and could be described by a compressed exponential (*Figure 2G*, black lines; *Figure 2—figure supplement 1A,B*; Appendix, Sections 9). When the viscoelastic parameters in our model were asymmetric, the shortening dynamics closely recapitulated this compressed exponential relaxation (*Figure 2G*, gray lines, I; expanded view in *Figure 2—figure supplement 1A,B*; Appendix, Sections 8–10).

Finally, we also analyzed junction length dynamics in non-shortening junctions in vivo (*Figure 2D, F*), because unlike shortening junctions, these display symmetrical mechanics along their length (*Figure 1E*, black, pink). Defining non-shortening junctions as any that displayed no net reduction in length over the observation time scale of ~400 s, we found that the length dynamics of non-shortening junctions in vivo displayed wide fluctuations over time. Moreover, non-shortening junctions did not share a self-similar relaxation pattern and displayed large deviations from the compressed exponential (*Figure 2H*, black, I; Appendix, Sections 8–10). Likewise, when symmetric viscoelastic

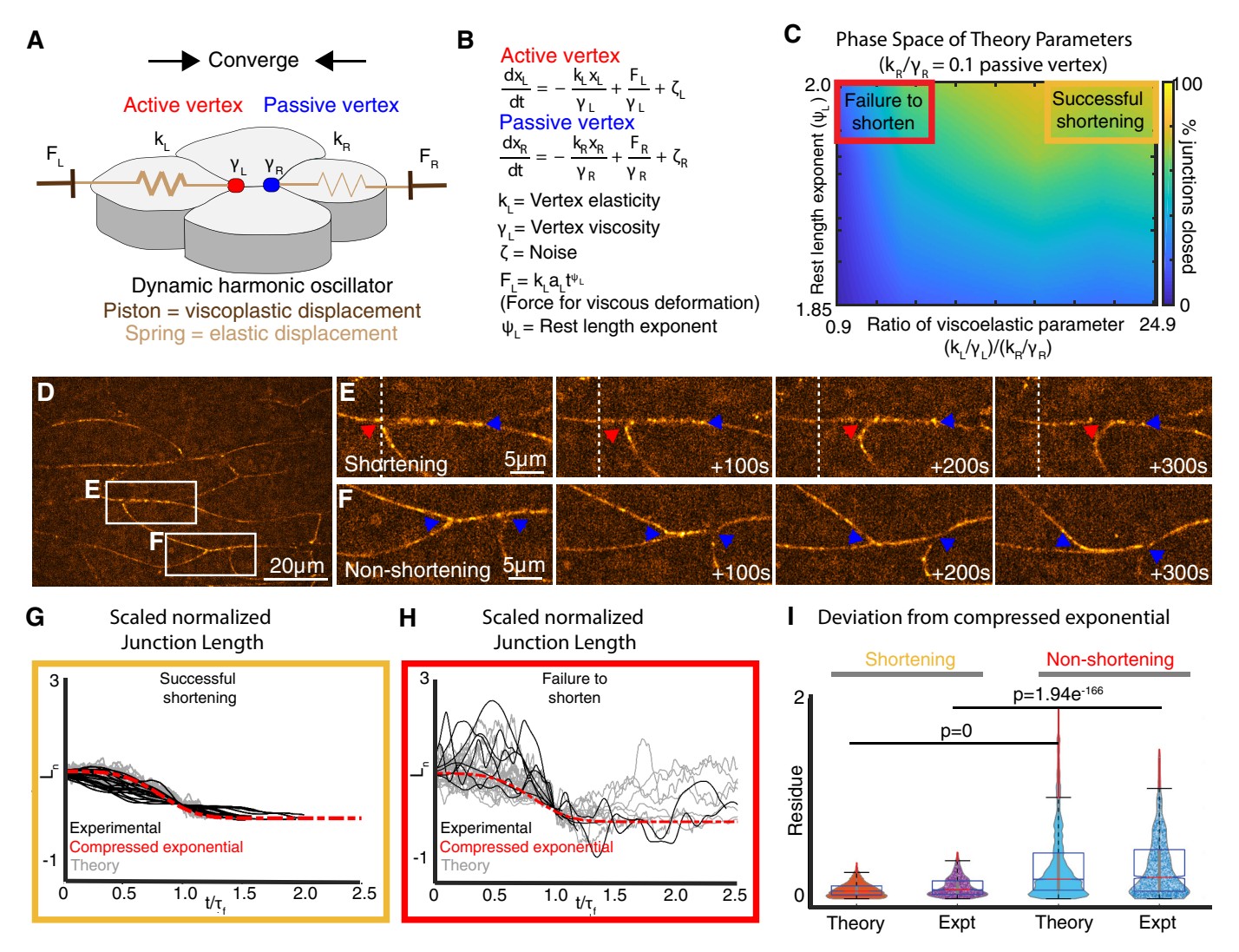

**Figure 2.** A new vertex model incorporating local mechanical heterogeneity recapitulates the fine-scale dynamics of junction shortening observed in vivo. (A) Sketch of v- junction shortening with elements of the model overlain. Active (red) and passive (blue) vertex movements are affected by a piston modulating the dynamic rest length. The vertices execute elastic motion due to springs of elasticity, $k_L$ and $k_R$. $L, R$ indices indicate left and right. The thicker spring indicates a stiffer elasticity constant, $k_L$. (B) Equations of motion for active and passive vertex positions, $x_L$ and $x_R$. Displacement of the left (right) vertex due to the piston is determined by the forces $F_L(F_R)$ whose time dependence is determined by the rest length exponent, $\psi_L(\psi_R)$. The friction experienced by the left (right) vertices are modeled using $\gamma_L(\gamma_R)$. $\zeta_L$ is the colored noise term for the left vertex (Appendix, Section 4-6). (C) Heatmap indicating probability of successful junction shortening (legend at right) in parameter space for the viscoelastic parameter near vertices and the rest length exponent, staying within biologically reasonable values based on data from *Drosophila* (*Solon et al., 2009*; Appendix, Section 6). (D) Still image from a time-lapse of *Xenopus* CE. Insets indicate representative shortening and non-shortening junctions shown in Panels E and F (vertices indicated by arrowheads). (G) Normalized change in length, $L_n$, for shortening junctions in vivo (black lines) and in simulations using asymmetric viscoelastic parameters (gray lines) resembling the compressed exponential form (red, dashed line) after the time axis is rescaled. (H) Normalized change in length, $L_n$, for non-shortening junctions in vivo (black lines) and in simulations using symmetrical viscoelastic parameters (gray lines). (I) Quantification of relaxation behavior deviation from the compressed exponential using the residue (Appendix, Section 8-10).

The online version of this article includes the following figure supplement(s) for figure 2:

**Figure supplement 1.** Extended analysis comparing in vivo and in silico junction dynamics.

parameters $(k/\gamma)$ were input into the model for both vertices bounding a single junction, the resulting junction length dynamics displayed wide fluctuations in length and deviated substantially from the normal relaxation pattern (*Figure 2H*, gray; I; Appendix, Sections 8–10).

Thus, by incorporating local mechanical heterogeneity, our new model not only recapitulates overall shortening/non-shortening outcomes, but also quantitatively recapitulates the dynamic patterns of length change observed in both shortening and non-shortening junctions in vivo. Because both modeling and observations suggest a key role for mechanical heterogeneity, we next sought to understand the contribution of such local mechanical regimes to cell movement during CE.

## Fluid-like directed motion of active vertices results from restriction of transverse fluctuations in motion

Our theory makes a prediction: that the more fluid-like motion of the active vertex occurs in the context of *increased* local stiffness (i.e. higher viscoelastic parameter), while the more glass-like motion of the passive vertex occurs in a relatively *decreased* stiffness regime. Given that vertex movement, while highly directional, is not *entirely* directed along the line joining two vertices (in-line movement) (*Figure 3A*), we reasoned that a stiffer mechanical regime might limit the movement of the active vertices perpendicular to the in-line direction; the perpendicular motion is referred to as transverse movement (*Figure 3A*, green). Transverse movement is indeed limited for active vertices thereby resulting in more smoothly processive, fluid-like motion in the line of shortening as compared to passive vertices (*Figure 3A*, orange).

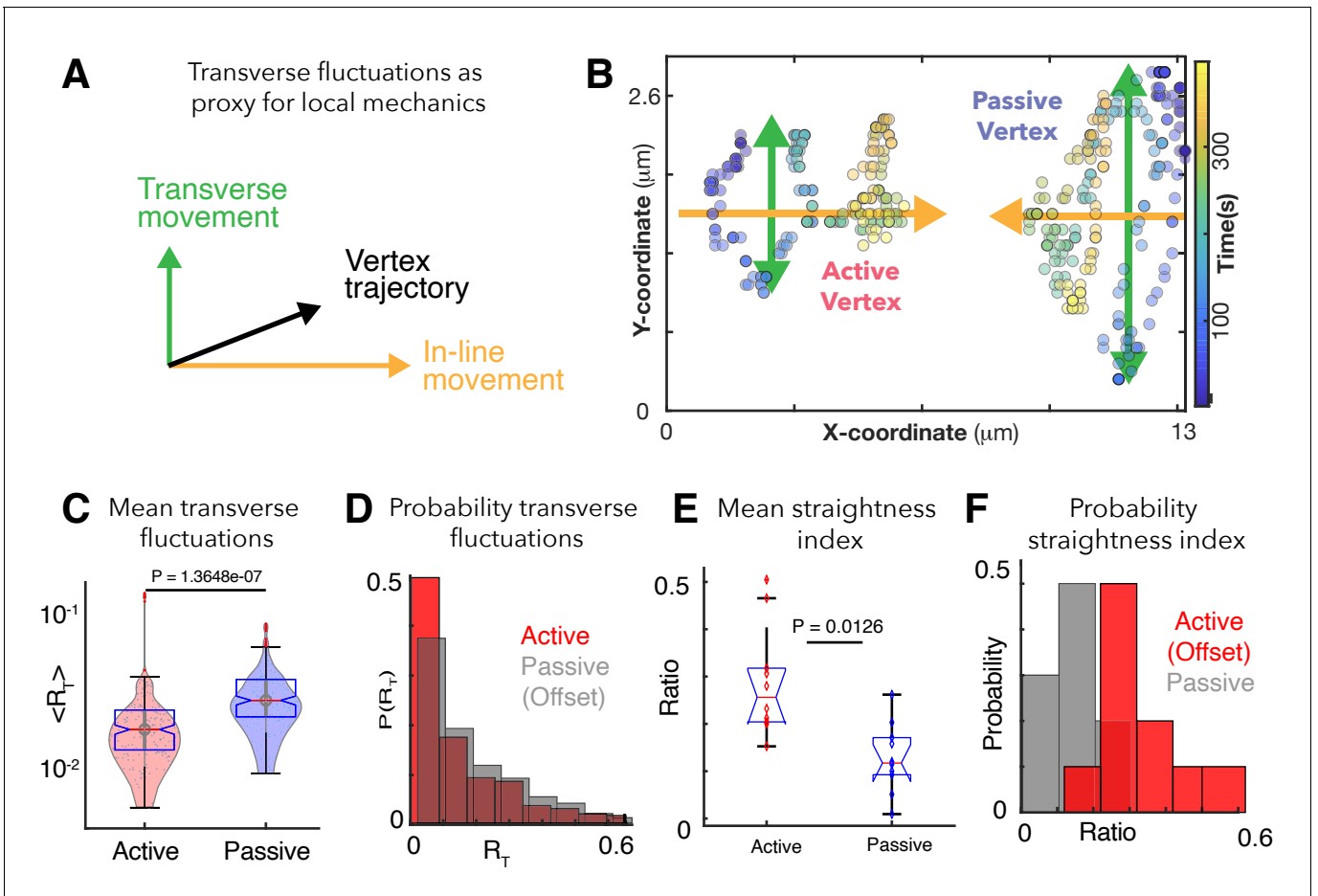

**Figure 3.** Patterns of transverse vertex fluctuations reveal mechanical heterogeneity of active and passive vertices in vivo. (**A**) Schematic of transverse fluctuations in the vertex position perpendicular to the direction of junction shortening; traverse movements are extracted using the transverse 'hop' function, which is inversely proportional to the local vertex stiffness (Appendix, Section 12). (**B**) X/Y coordinates for a representative pair of active and passive vertices color coded for time, with transverse (green) and in-line (orange) motion indicated. (**C**) Mean transverse fluctuation $\langle R_T \rangle$, for active and passive vertices. (N=20 vertices; 10 embryos over 386 seconds; t-test p value shown). (**D**) Probability distribution of transverse fluctuations, $R_T$, (offset for clarity). (**E**) Straightness index quantifying the persistence of vertex motion in terms of directionality (Appendix, Section 12); t-test p value is shown. (**F**) Probability distribution of the straightness index for active (red, offset for clarity) and passive (blue) vertices.

To test our model's prediction in vivo, we used our time-lapse data to quantify the transverse fluctuations of vertices (*Figure 3B*, green arrows; Appendix, Section 12). Consistent with our model's prediction, active vertices displayed significantly less transverse fluctuation than did passive vertices at the same junctions (*Figure 3B,C,E*), indicating a higher local stiffness at active vertices (*Marmottant et al., 2009*). Analysis of the straightness index, quantifying how straight vertices move along the in-line direction, independently validated this conclusion (*Figure 3E,F*; Appendix, Section 12).

This analysis of in vivo imaging data validates our physical model's prediction of an increased stiffness regime near active junctions and suggests that the lower stiffness regime of passive vertices allows more transverse fluctuation, resulting in less-directed, more glass-like movement. As such, multiple independent lines of observation and theory suggest that local mechanical heterogeneity along cell-cell junctions is a fundamental feature of CE. We next sought to understand the molecular underpinnings of this feature, asking if patterns of protein localization during CE might reflect the local mechanical patterns we identified here.

## Patterned *cis*-clustering of cadherins reflects the heterogeneous mechanics along shortening junctions

We first considered that the observed mechanical asymmetry along shortening v-junctions might result from asymmetric distribution of actomyosin, for example from asymmetric cellular protrusions from neighboring cells or asymmetric junction contraction events along v-junctions. We therefore measured actin intensity in the region abutting active and passive vertices, but we observed no such asymmetry (*Figure 4—figure supplement 1A–C*).

We next turned our attention to cadherin cell adhesion proteins, which have been shown to tune the very local mechanics of individual cell membranes in culture (*Strale et al., 2015*). We specifically examined Cdh3, as it is essential for CE in *Xenopus* (*Brieher and Gumbiner, 1994*; *Fagotto et al., 2013*; *Figure 4—figure supplement 1D*) and was recently implicated in CE cell movements in the mouse skin (*Cetera et al., 2018*). Like all classical cadherins, Cdh3 forms both intercellular *trans*-dimers and also *cis*-clusters mediated by intracellular interactions (*Figure 4A*; *Yap et al., 1997*). Such *cis*-clustering is a key regulatory nexus for cadherin function (*Yap et al., 2015*), so it is interesting that while the mechanisms governing *formation* of cadherin *cis*-clusters during CE has been studied, *cis*-cluster *function* during CE remains unknown (e.g. *Levayer and Lecuit, 2013*; *Truong Quang et al., 2013*).

We used high-speed super-resolution microscopy to image a functional GFP-fusion to Cdh3 and used the spatial autocorrelation function for an unbiased quantification of Cdh3-GFP cluster size (*Figure 4B*)(Appendix, Section 13). Using this function, an exponential decay in spatial correlation is expected for clusters that are regularly ordered, and this pattern was observed for Cdh3-GFP (*Figure 4C*). Moreover, this decay reached zero at ~1 μm (*Figure 4C*), consistent with the size reported for cis-clusters of cadherins in vertebrate cell culture (*Yap et al., 2015*).

This analysis revealed that together with pulsatile junction shortening (*Figure 4D*; *Shindo and Wallingford, 2014*) Cdh3 clusters undergo dynamic fluctuations in size (*Figure 4E*). Moreover, fluctuations in mean Cdh3 cluster size significantly cross-correlated with shortening pulses (*Figure 4F*). Mean cluster size peaked ~20 s prior to the onset of junction shortening pulses (*Figure 4E,F*)(Appendix, Section 14,15), suggesting a functional relationship between Cdh3 clustering and junction remodeling.

We then reasoned that mechanical heterogeneity observed along cell-cell junctions during CE might be driven by local patterns of Cdh3 clustering, since cadherins can tune the local mechanics of free cell membranes in single cultured cells (*Strale et al., 2015*). This led us to measure Cdh3 cluster size specifically in the ~3 micron region abutting vertices of shortening v-junctions during shortening pulses. Patterns of Cdh3 clustering were complex and highly heterogeneous, consistent with the mechanical heterogeneities we report here (*Figure 4—figure supplement 2*). Nonetheless, the mean size of Cdh3 clusters near active junctions was significantly larger than that for clusters near passive vertices (*Figure 4G*; Appendix, Section 16). We confirmed this important result using an alternative quantification of cluster size involving fits to the exponential decay of the spatial autocorrelation (*Figure 4—figure supplement 1E–G*; Appendix, Section 16).

Importantly, asymmetric Cdh3 clustering was specific to shortening v-junctions and was not observed along non-shortening junctions in the same tissue. Rather, all vertices bounding non-

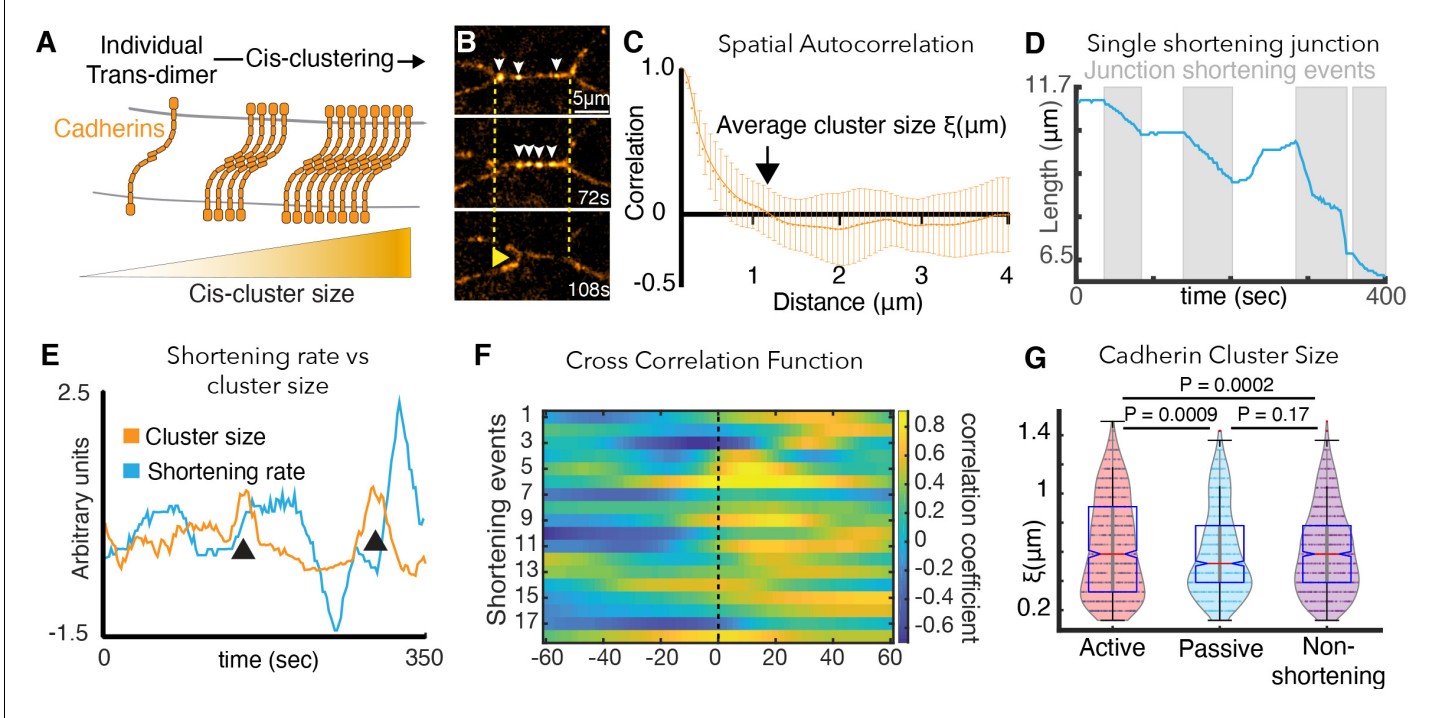

**Figure 4.** Cadherin cis-clustering correlates with vertex movements and mirrors asymmetric vertex dynamics. (**A**) C-cadherin (Cdh3) cis-clustering; trans-dimers form across opposing cell membranes (gray); lateral *cis* interactions drive clustering. (**B**) Frames from time-lapse of Cdh3-GFP; white arrows highlight clusters. Dashed lines denote initial vertex positions; yellow arrow indicates junction shortening. (**C**) Spatial autocorrelation of Cdh3 intensity fluctuations (SI Section 13)(60 image frames, 10 embryos). Autocorrelation decays to zero at ~1 μm. Error bars are standard deviation. (**D**) Trace from a single v-junction displaying pulsatile shortening highlighted by gray boxes (**E**) Junction length and Cdh3 cluster size fluctuations for an individual cell-cell junction. Cadherin cluster size fluctuations peak prior to junction shortening events (Appendix, Section 14,15). (**F**) Heat map showing cross correlation between junction length and Cdh3 cluster size. Color represents the value of the correlation coefficient (legend at right). Dashed black line indicates zero lag time. (Appendix, Section 14,15)(n = 11 junctions and 18 shortening events.) (**G**) Cadherin cluster size as extracted from spatial correlation curves (*Figure 4—figure supplement 2*; Appendix, Section 16). Cadherin cluster sizes are significantly larger near active vertices. Clusters near vertices of non-shortening junctions are not significantly different from those near passive vertices.

The online version of this article includes the following figure supplement(s) for figure 4:

**Figure supplement 1.** Extended analysis pertaining to cdh3 clustering and actin next to junctions (Appendix, Section 16).

**Figure supplement 2.** Source data for spatial correlation of Cdh3 intensity fluctuations reveal extreme heterogeneity in cluster size.

shortening junctions displayed clustering similar to that near passive vertices in shortening junctions (*Figure 4G*). Symmetrical clustering in non-shortening reflects the symmetrical dynamics of vertices bounding these junctions, described above (*Figure 1E*). Accordingly, these results demonstrate that asymmetric *cis*-clustering of Cdh3 is a specific property of shortening v-junctions during CE and suggests that such clustering may drive the asymmetric mechanics of active and passive vertices that we observed in vivo and predicted in silico.

## Cdh3 cis-clustering is required for axis elongation but not homeostatic tissue integrity in vivo

The patterned, asymmetric cis-clustering of Cdh3 during CE is a significant finding, because as mentioned above the *function* of cis-clustering remains undefined not only for CE, but indeed in any in vivo context. We therefore took advantage of point mutations in Cdh3 that specifically disrupt the hydrophobic pocket that mediates *cis* clustering, without affecting *trans* dimerization (cisMut-Cdh3; *Figure 5A*; *Harrison et al., 2011*; *Strale et al., 2015*). To test this mutant in vivo, we depleted endogenous Cdh3 as previously described (*Figure 5—figure supplement 1*; *Ninomiya et al., 2012*), and then re-expressed either wild-type Cdh3-GFP or cisMutant-Cdh3-GFP.

We first confirmed the *cis* mutant's impact on clustering in vivo. Re-expressed wild-type Cdh3-GFP clustered normally and displayed the expected exponential decay in spatial autocorrelation that

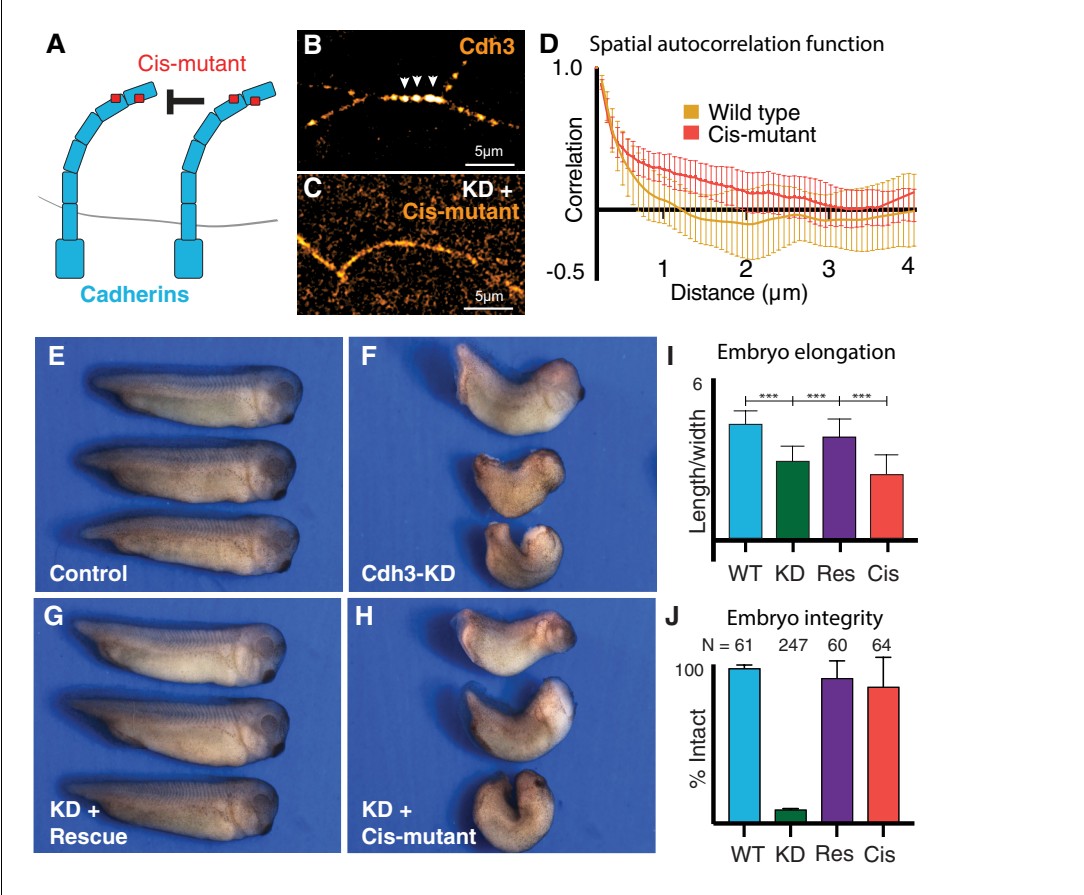

**Figure 5.** Cdh3 cis-clustering is required for convergent extension but not homeostatic tissue integrity. (**A**) Mutations used to inhibit cadherin cis-clustering. (**B**) Cdh3-GFP clustering in a control embryo. (**C**) Cis-clusters absent after re-expression of cisMut-Cdh3-GFP. (**D**) Mean spatial autocorrelation of Cdh3-GFP intensity fluctuations for wild type (60 image frames, from 10 embryos) and the cis-mutant (56 image frames, five embryos) (Appendix, Section 17). Gradual, non-exponential decay for cisMut-Cdh3-GFP indicates a lack of spatial order (i.e. failure to cluster). (**E**) Control embryos (~stage 33). (**F**) Sibling embryos after Cdh3 knockdown. (**G**) Knockdown embryos re-expressing wild-type Cdh3-GFP. (**H**) Knockdown embryos re-expressing cisMut-Cdh3-GFP. (**I**) Axis elongation assessed as the ratio of anteroposterior to dorsoventral length at the widest point. (**J**) Embryo integrity assessed as percent of embryos alive and intact at stage 23.

The online version of this article includes the following figure supplement(s) for figure 5:

**Figure supplement 1.** Cdh3 knockdown.

indicates regular spatial order and a mean cluster size ~1 µm (*Figure 5B,D*) (Appendix, Section 13,17). By contrast, when cisMut-Cdh3-GFP was re-expressed, clusters were clearly absent, and the signal was diffuse along cell-cell junctions (*Figure 5C*). Moreover, the spatial autocorrelation of cis-Mut-Cdh3-GFP did not decay exponentially (*Figure 5D*), consistent with a lack of spatial order (Appendix, Section 13,17). We confirmed this result using fits to the exponential decay of the spatial autocorrelation (*Figure 7—figure supplement 1*) (Appendix, Section 13,17).

We next used the same replacement strategy to directly test the function of cis-clustering in *Xenopus* CE. At neurulation stages, embryos depleted of Cdh3 display severe defects in axis elongation (*Figure 5E,F,I*, green) (*Brieher and Gumbiner, 1994*; *Lee and Gumbiner, 1995*). At later stages, these embryos disassociate to individual cells due to the widespread requirement for Cdh3 in cell cohesion (*Ninomiya et al., 2012*; *Figure 5J*, green). We found that re-expression of wild-type Cdh3-GFP rescued both axis elongation and embryo integrity, as expected (*Figure 5G,I,J*, purple).

Strikingly however, while re-expression of *cis*Mut-Cdh3-GFP significantly rescued overall embryo integrity (*Figure 5J*, red), it failed to rescue axis elongation (*Figure 5H,I*, red). These data provide the first experimental test of the role of cadherin cis-clustering in vivo, and moreover, provide an

experimental entry point for testing the role of cis-clustering in the generation of local mechanical patterns along cell-cell junctions.

## Loss of Cdh3 cis-clustering eliminates mechanical heterogeneity and disrupts shortening dynamics of cell-cell junctions during CE

To understand the relationship between Cdh3 clustering (*Figure 4*) and the asymmetric mechanics and vertex dynamics of shortening v-junctions (*Figures 1–3*), we applied our battery of physical methods to quantify the motion of vertices in cells with disrupted Cdh3 *cis*-clustering (i.e. Cdh3 knockdown +cisMut-Cdh3 re-expression). We found first that defects in axis elongation in cisMut-Cdh3 expressing cells were accompanied by defects in cell polarization (*Figure 6A–C*), reflecting the phenotype seen when PCP signaling is disrupted (*Wallingford et al., 2000*). Second, v-junctions in cells with disrupted Cdh3 clustering displayed large fluctuations in length that deviated significantly from the compressed exponential relaxation pattern observed for normal v-junctions (*Figure 6D,E*). The aberrant length dynamics of cisMut-Cdh3 expressing junctions resembled those of junctions that lack mechanical heterogeneity (i.e. non-shortening junctions in normal embryos in vivo or those modeled in silico (compare *Figure 6D* with *Figure 2H*)).

We then asked if cisMut-Cdh3 expression also disrupted the normal mechanical heterogeneity of v-junctions by quantifying transverse fluctuations of vertices. We found that all vertices in cells with defective cis-clustering of Cdh3 displayed the elevated transverse fluctuations observed only in passive vertices of normal cells (*Figure 6F,G*). These results provide direct experimental evidence that Cdh3 *cis*-clustering restricts transverse movement of vertices, thereby facilitating fluid-like shortening of the junction.

## PCP is essential for Cdh3 cis-clustering and mechanical heterogeneity at cell-cell junctions

A key challenge in animal morphogenesis is to understand how ubiquitous cellular machinery such as cadherin adhesion is directed by tissue-specific developmental control mechanisms. PCP signaling is a central regulator of vertebrate CE and PCP proteins localize to shortening v-junctions during *Xenopus* CE (*Figure 7A*), where they control actomyosin contractility (*Butler and Wallingford, 2018*; *Shindo et al., 2019*), but how these systems interface with cadherin adhesion during CE is poorly defined. Because cells with disrupted Cdh3 cis-clustering superficially resemble those with defective PCP (*Figure 6B,C*), we asked if Cdh3 clustering may be under the control of PCP signaling.

We disrupted PCP with the well-characterized dominant-negative version of Dvl2, Xdd1, which severely disrupted cell intercalation behaviors as expected (*Wallingford et al., 2000*; *Figure 6C*). Strikingly, expression of Xdd1 also elicited a significant disruption of Cdh3 clustering that was apparent in both images and in the lack of exponential decay in spatial autocorrelation data (*Figure 7B, C*). Finally, Xdd1 expressing junctions also displayed exaggerated length fluctuations, significant deviation from the compressed exponential relaxation behavior, and symmetrical, elevated transverse fluctuations (*Figure 7D–F*), all features associated only with junctions lacking local mechanical heterogeneity in vivo or in silico.

These data not only provide an independent experimental confirmation of the link between Cdh3 *cis*-clustering, local mechanical heterogeneity of junctions, and asymmetric vertex dynamics (*Figure 7G*), but also provide a novel mechanistic link between a conserved and essential developmental regulatory module (PCP), and the ubiquitous machinery of Cadherin adhesion.

## Discussion

Here, we combined physical and cell biological approaches to observation, theory, and experiment to identify and link two novel features of vertebrate convergent extension, one physical, the other molecular. First, we show that single cell-cell junctions in vivo display patterned mechanical heterogeneities along their length. Second, we show that locally patterned *cis*-clustering of a classical cadherin impart these patterns of mechanical heterogeneity under the control of PCP signaling.

These results are fundamentally important, because mechanical homeostasis in tissues is an emergent property of forces interacting across a wide range of length scales, yet we still know little about the subcellular mechanical properties of cells within tissues. Thus, while previous studies describe local heterogeneity in the membranes of single cultured cells (e.g. *Lieber et al., 2015*; *Shi et al.,*

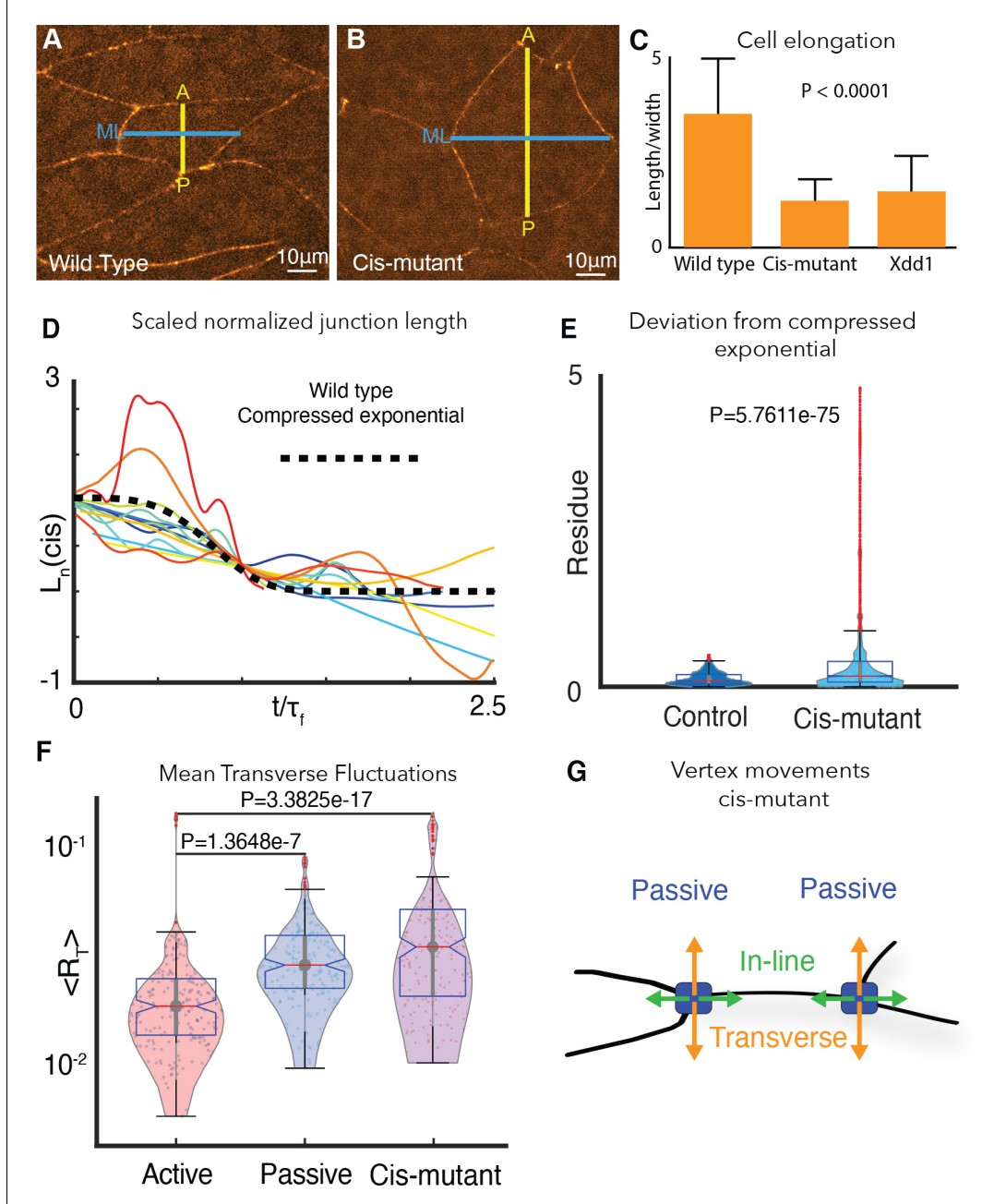

**Figure 6.** Cdh3 cis-clustering is required for heterogeneous junction mechanics. (**A**) Image of polarized, elongated control *Xenopus* mesoderm cells. Blue = mediolateral (ML); yellow = anterior-posterior (AP). (**B**) Stage-matched cells after depletion of endogenous Cdh3 and re-expression of cisMut-Cdh3. (**C**) Cellular length/width ratio to quantify CE cell behaviors (p value indicates ANOVA result). (**D**) Normalized junction length dynamics ($L_n$) for cis-mutant expressing junctions. Large fluctuations here are similar to those seen normally in non-shortening junctions (see *Figure 2H*). Dashed black line indicates the expected compressed exponential. (**E**) The residue quantifying significant $L_n$ deviation from the compressed exponential function as compared to control junctions. (**F**) Plots for transverse fluctuations $\langle R_T \rangle$, for control active and passive vertices compared to cis-mutant vertices. (Note: Data for active and passive junctions are re-presented from *Figure 3C* for comparison.) (**G**) Schematic illustrating symmetrical vertex behavior after disruption of cdh3 cis-clustering.

*2018*; *Strale et al., 2015*), our demonstration of local mechanical heterogeneity along single cell-cell junctions in an intact tissue is a substantial advance. Indeed, our data demonstrate that it is not the local heterogeneity per se, but rather its *local patterning* along individual cell-cell junctions that is a specific and essential feature of the junctional remodeling that drives CE. From a physical

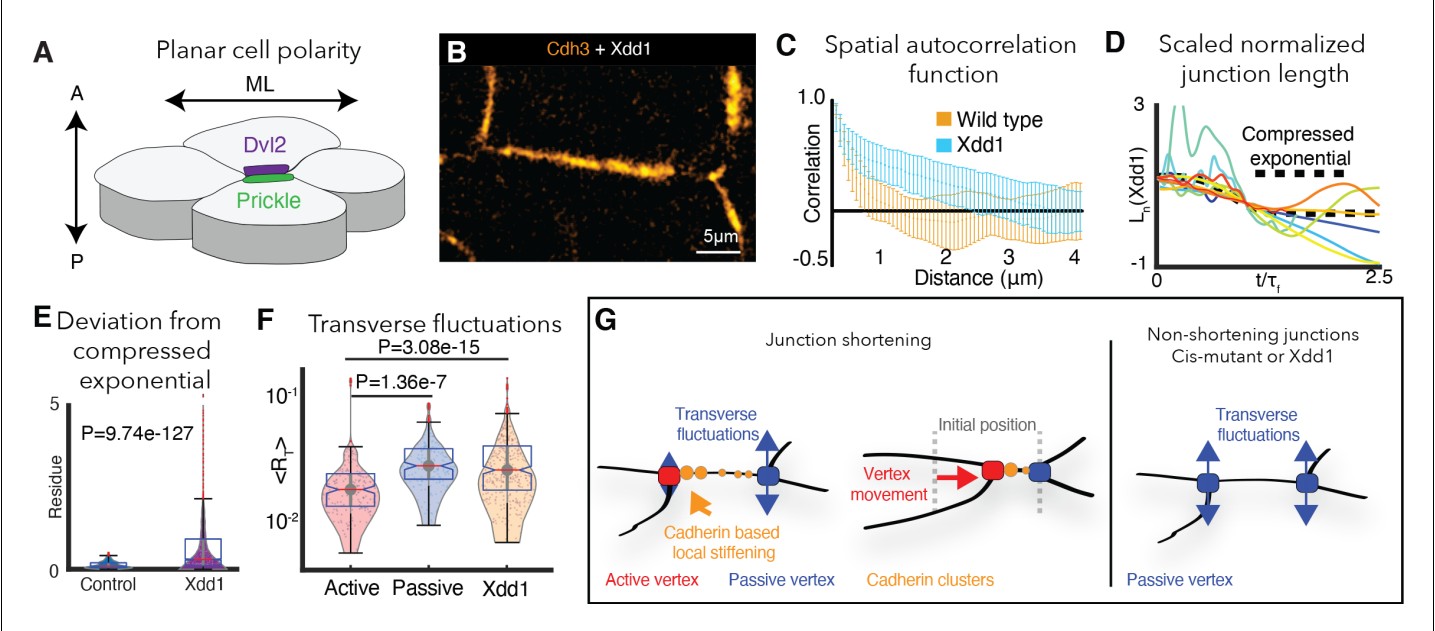

**Figure 7.** PCP is required for cdh3 cis-clustering and heterogeneous junction mechanics. (**A**) Cartoon of polarized core PCP protein localization. (**B**) Still image of Cdh3-GFP after expression of dominant negative Dvl2 (Xdd1). (**C**) Spatial autocorrelation of Cdh3 intensity fluctuations for Xdd1 (53 image frames, 5 embryos) and control embryos (60 frames, from 10 embryos),± std. dev. The spatial organization of Xdd1 mutant cadherin is similar to cisMut-Cdh3 expressing embryos. (**D**) Normalized junction length dynamics for Xdd1 embryos. Dashed black line indicates the normal compressed exponential behavior. (**E**) Residue for the deviation from the universal compressed exponential function for Xdd1 junctions. (**F**) Plots for transverse fluctuations at active and passive vertices compared to Xdd1-expressing vertices. (Note: Data for active and passive junctions are re-presented from *Figure 3C* for comparison to Xdd1.) (**G**) Schematic summarizing the primary conclusions.

The online version of this article includes the following figure supplement(s) for figure 7:

**Figure supplement 1.** Extended analysis of cadherin clustering for the cis-mutant, rescue, and Xdd1.

standpoint, this insight is important because it implies that the origin of patterned dynamic hetero-geneities observed at tissue length scales (e.g. *Angelini et al., 2011*; *Bi et al., 2015*; *Malmi-Kakkada et al., 2018*) may reside in similarly complex patterns at length scales as small as that of individual cadherin clusters.

Our findings are also important for understanding the unifying suite of CE cell behaviors that is deeply conserved across evolution. V-junction shortening is accomplished by a combination of cell crawling via mediolaterally positioned lamellipodia and active contraction of anteroposteriorly positioned cell-cell junctions (*Sun et al., 2017*; *Williams et al., 2014*), a pattern that has now been described in animals ranging from nematodes, to insects to vertebrates (*Huebner and Wallingford, 2018*). Although it remains to be determined whether v-junction short-ening in other tissues and animals also displays the heterogeneity we report here, it is nonetheless remarkable that even subtle aspects (e.g. active and passive vertices) are similar in tissues as diverse as *Drosophila* epithelial cells (*Vanderleest et al., 2018*) and *Xenopus* mesenchymal cells (*Figure 1*). Strikingly, a new preprint reports that asymmetric vertex behaviors are also observed when junction shortening is driven artificially in cultured cells by optogenetic activation of RhoA (*Cavanaugh et al., 2021*). Moreover, using an entirely independent modeling approach to the one we describe here, that work also suggests that local mechanical heterogeneity in cell-cell junctions is a fundamental feature of cell intercalation (*Cavanaugh et al., 2021*). Thus, asymmetric junction shortening may have a fundamental physical basis and is clearly ripe for further study.

Perhaps most importantly, our findings also have important implications for cadherin biology. The lateral cis-clustering of cadherins was first described decades ago (*Yap et al., 1997*) and has been extensively characterized using structural, biochemical, and cell biological approaches (*Chen et al., 2015*; *Fagotto et al., 2013*; *Hong et al., 2013*; *Kale et al., 2018*; *Levayer and Lecuit, 2013*; *Levayer et al., 2011*; *Truong Quang et al., 2013*; *Yap et al., 1998*). Because cadherin clustering is

thought to be driven by actomyosin contraction (*Yap et al., 2015*), our finding that PCP signaling is required for normal Cdh3 clustering is important. PCP proteins are enriched at shortening v-junctions, where they control pulsatile actomyosin contractions (*Butler and Wallingford, 2018*; *Shindo et al., 2019*; *Shindo and Wallingford, 2014*). It is reasonable, then, to posit that PCP-dependent actomyosin contraction is the key driver of Cdh3 clustering. However, another PCP protein, Frizzled7, has also been shown to tune cadherin adhesion during *Xenopus* CE, but bi-fluorescence complementation experiments suggested Frizzled7 *inhibits* cis-clustering (*Kraft et al., 2012*). Adding additional complexity is the recent finding that the PCP protein Celsr1, itself an atypical cadherin, not only forms adhesive trans-dimers but also tunes PCP signaling (*Stahley et al., 2021*). Unraveling the relationship between PCP, cadherins, and cell adhesion is thus an important challenge for future work.

Furthermore, despite the substantial body of work exploring the mechanisms by which cadherin clusters are formed, the *functional consequences* of defective cis-clustering in morphogenesis, or indeed in any intact tissue, have never been described. Our work therefore fills a critical gap and will be relevant far beyond the context of *Xenopus* axis elongation; for example Cdh3 (aka p-cadherin) is also implicated in PCP-mediated CE movements in the mouse skin (*Cetera et al., 2018*).

Our data argue that local asymmetric cis-clustering of Cdh3 is essential for the shortening of cell-cell junctions joining anteroposteriorly neighboring cells. Ultimately, such shortening is the result of the combined action of junction contraction and directed cell crawling (*Huebner and Wallingford, 2018*). In epithelial cells, contraction occurs apically, while cell crawling acts basolaterally (*Sun et al., 2017*; *Williams et al., 2014*). In *Xenopus* mesenchymal cells, which lack apical-basal polarity, we have found that the two mechanisms are integrated, driving more effective intercalation when both mechanisms act simultaneously (*Weng et al., 2021*). Another key challenge, therefore, will be to ask how the asymmetry of Cdh3 clustering observed here relates to contraction and/or cell crawling-based intercalation. Our imaging of actin dynamics argues against the simple interpretation that enhanced clustering relates directly to protrusive activity (*Figure 4—figure supplement 1*). Thus, integrating our findings here with previous work on Cdh3 in lamellipodial protrusions and in tissue boundary formation during *Xenopus* CE will also be important (see *Fagotto et al., 2013*; *Pfister et al., 2016*).

Finally, we note that our work here provides an important complement to the already extensive literature on CE and cadherin function in *Drosophila*. This is important because unlike all vertebrate animals, PCP proteins are dispensable for CE in *Drosophila* (*Zallen and Wieschaus, 2004*). Since PCP-mediated CE is essential for neural tube closure and PCP genes are among the most well-defined genetic risk factors for human neural tube defects (*Butler and Wallingford, 2017*; *Wallingford et al., 2013*), our data provide insights that span from the fundamental physics of living cells, to the cell and developmental biology of vertebrate axis elongation, to the etiology of human birth defects.

## Materials and methods

### *Xenopus* embryo manipulations

Ovulation was induced by injection of adult female *Xenopus* with 600 units of human chorionic gonadotropin and animals were kept at 16˚C overnight. Eggs were acquired the following day by squeezing the ovulating females and eggs were fertilized in vitro. Eggs were dejellied in 3% cysteine (pH 8) 1.5 hr after fertilization and embryos were reared in 1/3X Marc's modified Ringer's (MMR) solution. For microinjection, embryos were placed in 2% ficoll in 1/3X MMR and then washed in 1/3X MMR after injection. Embryos were injected using a Parker's Picospritzer III with an MK1 manipulator. Embryos were injected in the dorsal blastomeres at the four cells stage targeting the presumptive dorsal marginal zone. Keller explants were excised at stage 10.25 in Steinberg's solution using eyelash hair tools.

### Morpholino, plasmids, antibody, and cloning

The Cdh3 morpholino had been previously described (*Ninomiya et al., 2012*) and was ordered from Gene Tools. Cdh3-GFP, (*Pfister et al., 2016*) lifeact-RFP, and membrane-BFP were made in pCS105 and Xdd1 was made in CS2myc (*Sokol, 1996*). Cdh3 antibody was ordered from Developmental

Studies Hybridoma Bank (catalog number 6B6). The Cdh3-cis-mutant was generated using the Q5 Site-Directed Mutagenesis Kit (NEB, catalog number A13282) and here we changed valine 259 to aspartic acid and isoleucine 353 to aspartic acid.

## Morpholino and mRNA microinjections

Capped mRNA was generated using the ThermoFisher SP6 mMessage mMachine kit (catalog number AM1340). mRNAs were injected at the following concentrations per blastomere, Membrane-BFP (100 pg), Cdh3-GFP for imaging (50 pg), Cdh3-GFP for rescue (300 pg), Cdh3-cis-mutant (300 pg), lifeact-RFP (100 pg), and Xdd1 (1 ng). Cdh3 morpholino was injected at a concentration of 10 ng per blastomere.

## Imaging *Xenopus* explants

Explants were mounted on fibronectin coated glass coverslips in either Steinberg's solution or Danilchik's for Amy solution. Experiments were repeated in the absence of fibronectin to ensure fibronectin did not confound results. Explants were incubated at room temperature for 4 hr or at 16°C overnight before imaging. Standard confocal images were acquired with either a Nikon A1R or a Zeiss LSM 700. Super-resolution images were acquired with a commercially available instantaneous structured illumination microscope (BioVision Technologies). Standard confocal time-lapse movies were acquired with a 20 s time interval and super resolution images were acquired with a 2 s time interval. All images were acquired at a z-depth of 5 µm above the coverslip to insure similar z-depth sampling between images.

## Measurement of Cdh3 intensity at cell junctions

All image analyses were performed using the open-source image analysis software Fiji (*Schindelin et al., 2012*). Images were first processed with 50-pixel rolling ball radius background subtraction and smoothed with a 3 × 3 averaging filter, which allowed better distinction of individual cadherin clusters. The segmented line tool, with width set to the thickness of the junction (~16 pixels), was used to set a line of interest (LOI) across the length of the cell junction. Next the multi-plot tool was used to extract cdh3 intensity values across the length of the cell junction and the measure tool was used to collect data such as junction length and mean intensity values. The Fiji Time Lapse plugin Line Interpolator Tool was used to make successive measurements for movies. Here a segmented line LOI was drawn every 10–30 frames, the line interpolator tool was then used to fill in the LOIs between the manually drawn LOIs allowing rapid semi-manual segmentation. The multi-plot tool and measure tool were then used to extract data for each time-point of the movie. Source data for all imaging experiments can be found in the Dryad Server (doi: 10.5061/dryad.pg4f4qrph).

## Cdh3 immunostaining

Samples were prepared by micro-dissection as described above and incubated at room temperature for 4 hr or overnight at 16°C. Samples were then fixed in 1x MEMFA for 1 hr at room temperature and washed three times with PBS to remove fixative. Next samples were permeabilized with 0.05% Triton X-100 in PBS for 30 min and then blocked in 1% normal goat serum (NGS) in PBS for 2 hr at room temperature. The primary antibody was then diluted 1:100 in fresh 0.1% NGS/PBS and samples were incubated with primary antibody at 4°C overnight. Samples were then blocked a second time at room temperature for 1 hr and then washed twice with fresh blocking solution. Secondary antibody (goat anti-Mouse 488, #A32723) was diluted 1:500 and samples were incubated at 4°C overnight. Finally, samples were washed three times in 1X PBS and imaged.

## Embryo length to width measurement

Embryos were injected in the dorsal blastomeres with Cdh3-MO, Cdh3-MO + Cdh3 GFP (rescue), Cdh3-MO + Cdh3 cis-mutant (mutant),or left as un-injected controls. Live embryos were kept at room temperature for 26 hr post fertilization (~stage 33). Embryos were then fixed with MEMFA in glass vials on and rotated for 1 hr at room temperature. Post fixation samples were washed three times in 0.1% Tween-20 in 1X PBS and then images of embryos were acquired using a Zeiss AXIO Zoom stereoscope. The embryos anterior-posterior length and dorsal-ventral width were then measured using Fiji.

## Embryo survivability assay

Embryos were injected in the same manner as done for the length to width measurement and the number of embryos injected for each condition was recorded. Embryos were then kept at room temperature for 20 hr (~stage 20) and the number of surviving embryos was recorded. The percentage of embryos surviving (embryo integrity) was reported.

## Measurement of Cdh3 knockdown efficiency

Embryos were injected at the four-cell stage in a single dorsal blastomere with Cdh3-MO + membrane BFP generating embryos with mosaic knockdown of Cdh3 on the dorsal side of the embryo. Explants were next dissected from embryos, immuno-stained for Cdh3, and images were acquired as described above. The mosaic labeling allowed us to compare wild type and Cdh3-KD cells (marked by membrane-BFP) within a single explant. First, we used Fiji to measure endogenous Cdh3 intensity at cell junctions in wild type and Cdh3-KD cells and used a t-test to statistically compare these conditions. Next cellular polarity was assessed for each condition by measuring the ratio of the mediolateral length to the anterior-posterior width of individual cells.

## Acknowledgements

We thank Dan Dickinson for use of the iSIM microscope and for critical reading and helpful discussions. We thank Andy Ewald for critical reading of the manuscript. This work was supported by grants from the NICHD (R21HD084072) and the NIGMS (R01GM104853) to JBW and from the NSF (Phys 17–08128) and the Collie-Welch Chair through the Welch Foundation (F-0059) to DT.

## Additional information

### Funding

| Funder | Grant reference number | Author |
| --- | --- | --- |
| Eunice Kennedy Shriver National Institute of Child Health and Human Development | R01HD099191 | John B Wallingford |
| National Science Foundation | Phys 17-08128 | D Thirumalai |
| Welch Foundation | F-0059 | D Thirumalai |

The funders had no role in study design, data collection and interpretation, or the decision to submit the work for publication.

### Author contributions

Robert J Huebner, Conceptualization, Data curation, Formal analysis, Validation, Investigation, Visualization, Writing - review and editing; Abdul Naseer Malmi-Kakkada, Conceptualization, Data curation, Software, Formal analysis, Visualization, Methodology, Writing - review and editing; Sena Sarıkaya, Investigation, Visualization; Shinuo Weng, Investigation, Methodology; D Thirumalai, Formal analysis, Supervision, Funding acquisition, Visualization, Project administration, Writing - review and editing; John B Wallingford, Conceptualization, Supervision, Funding acquisition, Writing - original draft, Project administration, Writing - review and editing

### Author ORCIDs

Robert J Huebner ⓘ https://orcid.org/0000-0001-8778-9689
Abdul Naseer Malmi-Kakkada ⓘ https://orcid.org/0000-0002-5429-4652
Sena Sarıkaya ⓘ https://orcid.org/0000-0001-5008-2065
Shinuo Weng ⓘ https://orcid.org/0000-0001-7932-913X
D Thirumalai ⓘ https://orcid.org/0000-0003-1801-5924
John B Wallingford ⓘ https://orcid.org/0000-0002-6280-8625

## Ethics

Animal experimentation: Animal work described here was performed in accordance with the UT Austin Institutional Animal Care and Use Committee protocol #AUP-2018-00225.

## Decision letter and Author response

Decision letter https://doi.org/10.7554/eLife.65390.sa1
Author response https://doi.org/10.7554/eLife.65390.sa2

# Additional files

## Supplementary files

• Transparent reporting form

## Data availability

Raw data from time-lapse imaging are available on Dryad.

The following dataset was generated:

| Author(s) | Year | Dataset title | Dataset URL | Database and Identifier |
|---|---|---|---|---|
| Huebner RJ, Malmi-Kakkada AN, Sarıkaya S, Weng S, Thirumalai D, Wallingford JB | 2021 | Data from:Mechanical heterogeneity along single cell-cell junctions is driven by lateral clustering of cadherins during vertebrate axis elongation | https://doi.org/10.5061/dryad.pg4f4qrph | Dryad Digital Repository, 10.5061/dryad.pg4f4qrph |

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

## Appendix 1

### Section 1. Active versus passive vertex dynamics

We used the Manual Tracking plugin in FIJI to obtain the trajectories of vertex pairs. Individual vertex positions were tracked for a time interval of 400 s every 2 s. By obtaining the time-dependent two-dimensional (2D) vertex co-ordinates $(x_L, y_L)$ and $(x_R, y_R)$ for the left (L) and right (R) vertices respectively, the net distance travelled by the left(L) vertex is,

$$\Delta r_L = \sqrt{(x_L(t_f) - x_L(t_0))^2 + (y_L(t_f) - y_L(t_0))^2}, \tag{1}$$

where $x_L(t_f)$, $x_L(t_0)$ are the vertex positions at the final $(t_f)$ and initial time $(t_o)$ of measurement respectively. A similar equation with $x_R, y_R$ applies for the right vertex. The length of the junction is,

$$L(t) = \sqrt{(x_R(t) - x_L(t))^2 + (y_R(t) - y_L(t))^2} \tag{2}$$

To determine the weight of the contribution of each vertex to junction shortening, we define an activity parameter, $A$, as the ratio of net vertex distance moved to the initial junction length that is $A_L = \frac{\Delta r_L}{L(t_0)}$. Similarly, $A_R = \frac{\Delta r_R}{L(t_0)}$, for the right vertex. If $A_L > A_R$, the left vertex is labelled as the 'active' vertex while the right vertex is the 'passive' one, and vice versa if $A_R > A_L$. Over the time frames that we have analyzed the vertex movement, the median value of $L(t_f)/L(t_0) \sim 0.30$, implying that the junctions have shortened by $\sim 70\%$ as compared to the initial junction length. Both high time resolution (2s per frame) and low time resolution (20s per frame) imaging data show the same trend that one of the vertices tend to be active, contributing more to junction shortening (*Figure 1B–C*, Main Text). We confirm that this observation is not due to the overall motion of the cells as detailed below (*Figure 2—figure supplement 2*).

### Section 2. Quantifying the heterogenous dynamics of vertices: Mean Square Displacement (MSD), van Hove function, and the velocity autocorrelation

The characteristics of vertex dynamics could provide clues as to the active mechanisms that promote or impede vertex movement. An important parameter to quantify vertex dynamics is the Mean Square Displacement (MSD), as a function of the lag time t. Time averaged MSD, $\overline{\Delta}(t)_i$, is calculated using the vertex positions $\vec{r}_i(t')$,

$$\overline{\Delta}(t)_{L,i} = \frac{1}{T-t} \int_0^{T-t} \left| \vec{r}_{L,i}(t'+t) - \vec{r}_{L,i}(t') \right|^2 dt' \tag{3}$$

where $T = 400s$ and subscript $L$ stands for the left vertex. Taking the average over N independent vertex trajectories, labelled by the index $i$, we obtain the ensemble averaged MSD, $\Delta(t)_L = \frac{1}{N} \sum_{i=1}^{N} \overline{\Delta}(t)_{L,i}$.

The same procedure is used to calculate the MSD for the right vertex (see *Figure 1E*, Main Text). In many physical systems, the MSD increases with a power law, that is. $\Delta(t) \sim t^\alpha$. When the vertex motion is uncorrelated in time and along random directions, the dynamics is described as Brownian, and the MSD exponent is unity, $\alpha = 1$. Sub-diffusive, $\alpha < 1$, movement occurs when there is a hindrance to motion or the dynamics is highly correlated. For example, when a particle in caged by its immediate neighbors, sub-diffusive motion results. Super-diffusive MSD, $\alpha > 1$, is seen when the motion is highly directed.

We found substantial heterogeneity in the individual vertex MSD as seen from the plot of $\overline{\Delta}(t)_i$ (*Figure 14—figure supplement 2C*). Active and passive vertex MSDs span 3 orders of magnitude of time lag. Two distinct time regimes are observed for both active and passive vertex movements: (i) at short time lags, $t < 30s$, active and passive vertex movements are random, characterized by MSD exponent $\alpha \sim 1$. (ii) For $t > 40s$, active vertices show strong superdiffusive movement while passive vertices undergo a slowdown followed by a recovery toward superdiffusive motion (see

*Figure 1E,F* Main Text). These distinct differences between active versus passive vertices are observed in the ensemble averaged MSD, $\Delta(t)$ for 20 vertices from 10 different embryos.

To eliminate the effect of motion of the entire tissue, we tracked vertex positions with respect to the center of an egg yolk particle (*Figure 1—figure supplement 2D, E*) typically present within cells as well as nearby stationary vertices (*Figure 1—figure supplement 2F, G*). In this manner, we analyzed the relative vertex positions, $\vec{r}_{rel}$, with respect to a frame of reference within the tissue being imaged. By extracting the co-ordinates of the center of an egg yolk within a cell or nearby vertices, $\vec{r}_c$, we obtain the relative vertex positions, $\vec{r}_{rel,L} = \vec{r}_L - \vec{r}_c$. We then evaluated the mean square relative displacements (MSRD) for the left and right vertex pairs using *Equation (3)* above (*Figure 1—figure supplement 2E, G*). The distinct differences between active versus passive vertex dynamics is conserved in this relative co-ordinate system, indicating that the asymmetry in active versus passive vertex movement is not due to motion of the whole tissue (see *Figure 1—figure supplement 2D, G*).

## Van Hove function

Insights into vertex motion may be obtained by analogy to spatially heterogenous dynamics in supercooled liquids (*Barrat et al., 1990*; *Thirumalai and Mountain, 1993*). The distribution of particle displacements is expected to be a Gaussian in simple fluids. In supercooled liquids, however, the displacements of a subset of particles deviate from the Gaussian distribution (*Thirumalai and Mountain, 1993*). From the distance moved by a vertex during the time interval $\delta t$, defined as $|\delta r_i(\delta t)| = |\vec{r}_i(t + \delta t) - \vec{r}_i(t)|$, the van Hove function for vertex displacement (or the probability distribution of vertex step size) is,

$$P(\delta r | \delta t) = \left\langle \frac{1}{N} \sum_{i=1}^{N} \theta\left(\delta r_i - \left|\vec{r}_i(t + \delta t) - \vec{r}_i(t)\right|\right) \right\rangle \tag{4}$$

where the average is over N independent vertex trajectories. The van Hove distribution at $\delta t = 40s$, for active (red) and passive vertices (blue) is shown in (*Figure 1—figure supplement 3A*). The $40s$ time interval is long enough to clearly observe the differences in the distances moved by active and passive vertices. The van Hove distribution at $\delta t = 4s$ is shown in (*Figure 1—figure supplement 3B*). At this short time interval, distances moved by active and passive vertices are similar and is well fit by a Gaussian (see inset *Figure 1—figure supplement 3B*). However, the van Hove distribution deviates significantly from the Gaussian distribution at $\delta t = 40s$ (see inset *Figure 1—figure supplement 3A*), indicating the growing heterogeneity in the vertex displacements.

## Average velocity distribution and velocity autocorrelation function (VACF)

To further quantify the striking differences in the movement of active and passive vertices, we calculate the average velocity of the vertices. The average velocity over a time interval $\tau$ is defined as,

$$\vec{v}_L(\tau) = \frac{\vec{r}_L(t) - \vec{r}_L(t + \tau)}{\tau} \tag{5}$$

Replacing $\vec{r}_L$ by $\vec{r}_R$ gives the average velocity of the right vertex. We analyze the average velocity over a time interval $\tau$ because experimental data is also an average over the time resolution of the iSIM microscope. We then compare the speed distribution ($|\vec{v}_L(\tau)|$) of active and passive vertices over both short, $\tau = 4s$ (*Figure 1—figure supplement 3C*, blue for passive and red for active vertices) and longer time intervals, $\tau = 60s$ (*Figure 1—figure supplement 3D*). At the smaller time interval, $\tau = 4s$, the speed distribution of active and passive vertices are similar. This indicates minimal differences between active and passive vertex dynamics at short time scales. The difference in active and passive speed distribution is, however, pronounced at $\tau = 60s$. The passive vertex speed distribution peaks at a smaller value and decays rapidly for larger speed values, compared to active vertices. This illustrates the fluidization in the movement of active vertices that develops over a time scale of order $50s$ in agreement with other measures such as the MSD and the van Hove distribution as reported above.

To probe the time interval over which the average velocity (at fixed $\tau$) is correlated with average velocity at a time point separated by $\delta t$, we calculate the velocity autocorrelation function (VACF),

$$C_v^\tau(\delta t) = \left\langle \vec{v}(t + \delta t).\vec{v}(t) \right\rangle \tag{6}$$

where the average is defined as $\langle\ldots\rangle = \frac{1}{T - \delta t}\int_0^{T-\delta t}\ldots dt$. The VACF is normalized such that $C_v^\tau(\delta t = 0) = 1$. At the shorter time interval of $\tau = 4s$, VACF for active and passive vertices exhibit a rapid decay to zero (*Figure 1—figure supplement 3E*), blue for passive and red for active vertices. Individual vertex VACF are plotted in transparent colors and the mean as dashed lines (blue-black dashed line for passive vertices and red-black dashed line for active vertices).

Analyzing vertex velocities at $\tau = 60s$ clearly brings out the different dynamics that characterize active versus passive vertices (*Figure 1—figure supplement 3F*, blue for passive and red for active vertices). Velocity correlations decay quicker for passive vertices, becoming negative and then rebounds. However, active vertex velocity correlations are more persistent with time as evident from the longer time to decay.

## Section 3. Self-overlap parameter and dynamic heterogeneity

To quantify the highly asymmetric vertex movement that underlies CE, we measured the fractional change in vertex positions over a time interval t using the self-overlap order parameter, defined as:

$$\langle Q(t)\rangle = \frac{1}{N}\sum_i^N w_i \tag{7}$$

where $w_i = 1$ if $\left|\vec{r}_i(t' + t) - \vec{r}_i(t')\right| < L_c$ and $w_i = 0$ otherwise. The self-overlap parameter is dependent on the length scale that is probed by $L_c$ and represents the probability that vertices have moved by a specified length scale over a time interval, t. We chose $L_c = 1.3\mu m$, as this is the distance scale over which movement of active and passive vertex become distinct. This is evident from the plot of MSD (*Figure 1E,F*, Main Text) for active and passive vertices where the dynamics begins to differ at a length scale of $>1\mu m$. If a vertex moves less than $L_c = 1.3\mu m$ over the time interval t, the vertex is considered to have 100% overlap with its previous position, and hence assigned a value 1. However, if the vertex has moved more than $1.3\mu m$ within the time interval $t$, we consider this as 0% overlap. The self-overlap function, $\langle Q(t)\rangle$, is calculated by averaging over a range of initial times, $t'$, followed by ensemble averaging over individual vertices (*Figure 1—figure supplement 3G-H*). The active vertex self-overlap function decays rapidly and can be fit to a single exponential decay function, indicating liquid like dynamics. However, passive vertex overlap function shows a two-step decay, a signature of glass-like dynamics (*Figure 1—figure supplement 3G*).

Although the MSD and the self-overlap function $\langle Q(t)\rangle$ are useful to quantitatively characterize vertex movement, other metrics are needed to gather further insights into the dynamic heterogeneity and correlations in vertex movement that emerge temporally during CE. In systems approaching the glass transition, the cooperativity of motion increases such that the length and time scales characterizing the dynamic heterogeneity are expected to grow sharply. In supercooled liquids, the fourth order susceptibility, $\chi_4(t)$, provides a unique way to distinguish the dynamic fluctuations between liquid and frozen states (*Kirkpatrick and Thirumalai, 1988*). Therefore, we compute the fourth order susceptibility from the variance of the self-overlap parameter,

$$\chi_4(t) = \left\langle Q(t)^2 \right\rangle - \langle Q(t)\rangle^2 \tag{8}$$

Similar to structural glasses, the dynamic heterogeneity, quantified by $\chi_4(t)$ increases with time, peaks at a maximum time interval, $t_M$ and then decays (*Figure 1—figure supplement 3I*). The dynamic heterogeneity is manifested as dramatic variations between individual vertex trajectories in both active and passive vertex movements. For active vertices, $\chi_4(t)$ peaks at $t_M \sim 120s$ while for passive vertices heterogeneity peaks at a longer time interval $t_M \sim 170s$ (*Figure 1—figure supplement 3I*). The time scale associated with the peak in dynamic heterogeneity is consistent with the viscoelastic relaxation time (further discussed below), known to be the characteristic relaxation time for vertices connected by the cell cortex under tension (*Solon et al., 2009*). For non-shortening

junctions, $\chi_4(t)$, does not show a peak (*Figure 1—figure supplement 3J*). We anticipate the peak to be at a much longer time scale for vertices of non-shortening junctions.

## Section 4. Theoretical model

Vertex based models are important for studying the dynamics of confluent cell layers (*Fletcher et al., 2014*). The junction between three or more cells (vertices) are represented as point particles. The connecting edge between vertices represent cell-cell interfaces. We developed a theoretical model for junction shortening to understand the asymmetric dynamics of vertices. Our model, shown in (*Figure 2A,B* Main Text), is a coarse-grained representation of a collection of cells intercalating mediolaterally. Each vertex, bounding the v-junction, are connected to Maxwell-like components with viscous and elastic elements. Elastic properties are modeled by springs with stiffness, $k$, and actuators characterize the viscous motion of cell vertices (see *Figure 2A* Main Text; $\gamma$ is the viscosity). For the purposes of visualization, we depict the spring-actuator element as being in the direction away from the cell-cell interface, exerting a compressive force on the vertices. This need not be the case as the forces and mechanical factors contributing to junction shortening can also be localized within the cell-cell junction. For the purposes of simplicity in visualization, we picked a direction for the spring-actuation element.

We assume that the position of the left vertex, $\vec{r}_L(Noentity x_L, y_L)$, evolves according to the equation of motion:

$$\frac{d\vec{r}_L}{dt} = -\frac{\vec{k}_L \cdot \vec{r}_L}{\gamma_L} + \frac{\vec{F}_L}{\gamma_L} + \zeta_L \tag{9}$$

where $\vec{k}_L$ is the elasticity of the left ($L$) vertex, $\vec{F}_L$ is the contractile force responsible for viscous deformation of the vertex and $\gamma_L$ is viscosity coefficient of the vertex. Replacing the subscript $L$ with $R$ above gives the equation of motion for the right vertex. The local elasticity near the vertices are accounted for by a connected harmonic spring with strength $\vec{k}_L$. The spring is connected in series with an actuator that supplies the contractile force, $\vec{F}_L$. It is likely that the noise in a physical or biological system is correlated in time. Consistent with our observation that fluctuations in junction length are correlated in time, we model $\zeta_L$ as the colored noise experienced by the vertices. The noise, $\zeta_L$, represents the coupling of the vertices to their immediate local environment, satisfying $\langle \zeta_L(t)\zeta_L(s)\rangle = Ae^{-|t-s|/\tau_n}$ with the mean $\langle \zeta_L(t)\rangle = 0$. The coefficient, $A$, is the noise strength. For large noise strength, vertex positions show large amplitude deviations from the position dictated by the minimum of the elastic force, as constrained by the spring. For small persistence time of the correlated noise, $\tau_n$, the vertex dynamics is highly uncorrelated in time. At large persistence times, however, the noise induced fluctuations in the vertex positions are correlated over the timescale $\tau_n$. We set the noise correlation time to be the persistence time of junction length fluctuations. The colored noise satisfies, $\frac{d\zeta_L}{dt} = -\frac{\zeta_L}{\tau_n} + \frac{1}{\tau_n}\eta(t)$, where $\eta(t)$ is the Gaussian white noise source characterized by delta correlation $\langle \eta(t)\eta(s)\rangle = \delta(t-s)$ and mean $\langle \eta \rangle = 0$.

Since the movement of vertices along the medio-lateral direction is much more persistent as opposed to the perpendicular direction, as evident from the closure of junctions, we simplify the model to consider only one-dimensional (1D) motion. Henceforth, we drop the vector notation and focus on the vertex dynamics along the x-axis.

By considering the basic vertex equations in the Langevin picture,

$$\frac{dx_L}{dt} = -\frac{k_L}{\gamma_L} \times \left(x_L - a_L t^{\psi_L}\right) + \zeta_L \tag{10}$$

$$\frac{dx_R}{dt} = -\frac{k_R}{\gamma_R} \times \left(x_R - \left(L_0 - a_R t^{\psi_R}\right)\right) + \zeta_R \tag{11}$$

we model the vertex equations of motion in analogy to particles moving in a translating optical trap. The minimum of the left elastic 'trap' changes dynamically due to the term $a_L t^{\psi_L}$ in *Equation 10* (modeled by the left actuator). Similarly, the right elastic 'trap' is translated from its initial position $L_0$

by $a_R t^{\psi_R}$ in *Equation 11* (modeled by the right actuator). These terms serve as a proxy for active contractile forces which viscously deform the cell edges. Hence, we refer to the exponents, $\psi_L$ and $\psi_R$, as the rest length exponents. The physical implication of the rest length exponent is that the rest length of the junction varies dynamically. The contractile force is, $F_{L,total} = -V'(x_L)$ in *Equation 9*, where $V(x_L)$ is the time-dependent 'trap' potential of the form $V_L = 0.5 k_L \left( x_L - a_L t^{\psi_L} \right)^2$ and $V_R = 0.5 k_R \left( x_R - \left( L_0 - a_R t^{\psi_R} \right) \right)^2$. The stochastic movement of the vertices in a translating potential leads to a *ratchet-like* effect where the vertex dynamics has a specified direction. This directionality in the motion of the vertices does not arise, however, from the asymmetry in the potential but rather from the asymmetric translation of the potential well minimum or the dynamic rest length.

Hence, the active time-dependent forces contributing to junction shortening were modeled in silico as,

$$F_L(t) = k_L a_L t^{\psi_L} \tag{12}$$

$$F_R(t) = k_R \left( L_0 - a_R t^{\psi_R} \right) \tag{13}$$

where $a_L$ and $a_R$ are the 'acceleration' of the left and right vertices respectively, and the exponents $\psi_L$ and $\psi_R$ determine the temporal dynamics of the contractile force. We include the acceleration term to account for the experimentally observed increase in the persistence of junction shortening as a function of time (see *Figure 2G* Main Text; See also Sec. 9 below). The initial condition is set as $x_L(t=0) = 0$, and $x_R(t=0) = L_0$, with $L_0$ being the initial cell-cell junction length. We arbitrarily assign the left side to be active, with the time dependent active force rising in proportion to $t^{\psi_L}$ ($\psi_L > \psi_R$). The right side is assigned to be passive, with force increasing with time as $\propto t^{\psi_R}$. The difference in the rest length exponents, $\psi_L$ versus $\psi_R$, determines which vertex is active.

The equations of motion then become:

$$\frac{dx_L}{dt} = -\frac{k_L}{\gamma_L} x_L + \frac{k_L a_L t^{\psi_L}}{\gamma_L} + \zeta_L \tag{14}$$

$$\frac{dx_R}{dt} = -\frac{k_R}{\gamma_R} x_R + \frac{k_R \left( L_0 - a_R t^{\psi_R} \right)}{\gamma_R} + \zeta_R \tag{15}$$

Defining $\bar{x}_L = \frac{x_L}{x_0}$, $\bar{t} = \frac{t}{\tau}$ and $\bar{a}_L = \frac{a_L}{a_L^0}$, where $x_0 = 10\mu m$, $\tau = 10^2 sec$ and $a_L^0 \tau^{\psi_L} = x_0$, we recast the equations of motions into dimensionless forms. Similar normalization with L replaced by R applies for the right vertex. The system of equations is scaled with the characteristic length and time, $x_0$ and $\tau$, physiologically relevant for cells undergoing convergent extension. In terms of the normalized quantities, the equation of motion is,

$$\frac{d\bar{x}_L}{d\bar{t}} = -\frac{1}{\bar{\tau}_L} \times \left( \bar{x}_L - \bar{a}_L \bar{t}^{\psi_L} \right) + \bar{\zeta}_L \tag{16}$$

$$\frac{d\bar{x}_R}{d\bar{t}} = -\frac{1}{\bar{\tau}_R} \times \left( \bar{x}_R - \left( \bar{L}_0 - \bar{a}_R \bar{t}^{\psi_R} \right) \right) + \bar{\zeta}_R \tag{17}$$

where the parameter $\frac{k_L}{\gamma_L} = \frac{1}{\tau_L}$ has the dimension of inverse time $\frac{1}{s}$. When normalized by the characteristic timescale $\tau$, $\frac{\tau}{\tau_L} = \tau \times \left( \frac{k_L}{\gamma_L} \right) = \frac{1}{\bar{\tau}_L}$, we obtain a dimensionless parameter which we refer to as the viscoelastic ratio.

## Section 5. Dynamic rest length and colored noise

In vertex-based models for plant cells, the cell-cell interface length is modeled with a spring having a characteristic rest length (*Merks et al., 2011*). Any deviation in the length of the cell-cell interface from the rest length is energetically unfavorable. In vertex models for animal cells, such a rest length is typically not included (*Fletcher et al., 2014*). In our coarse-grained vertex model, we include a

spring term with dynamic rest lengths. We show that this model accounts for the asymmetric vertex dynamics and quantitative experimental features of the junction shortening behavior. By studying actomyosin contractility in combination with theoretical modeling, it has recently been shown that epithelial junctions exhibit both elastic and viscous remodeling behavior (*Staddon et al., 2019*).

The existence of memory effects in junction shortening necessitates the addition of the colored noise term. Previous vertex-based models have considered random white noise indicating no memory effect. However, by experimentally quantifying the junction length fluctuations, we would like to point out that colored noise may be important to consider in modeling biological systems.

## Section 6. Parameter values for elasticity and viscosity:

The viscoelastic ratios, $1/\overline{\tau}_L$ and $1/\overline{\tau}_R$, were varied from 0.05 to 5 equivalent to $5 \times 10^{-4}s^{-1} - 0.05s^{-1}$ in dimensional units. Therefore, the viscoelastic relaxation time is in the range of $20s - 2000s$. Spring stiffness, $k$, in the range between $100pN/\mu m$ and $1nN/\mu m$ (*Bittig et al., 2008*; *Girard et al., 2007*) and the viscosity, $\gamma \sim 100nN.s/\mu m$ (*Forgacs et al., 1998*), accounts for the elastic and viscous properties of tissues previously reported in the literature. For these values, one obtains the viscoelastic relaxation time in the range of $1s - 100s$. Therefore, the viscoelastic ratio used in our model is within an order of magnitude of the physiological values for both tissue stiffness and viscosity.

## Section 7. Simulation details

We consider a wide range of values for both the viscoelastic ratio and the rest length exponent for the active vertex, $\psi_L$. The time step in the simulation is $\Delta t = 0.0022 = 0.22s$, chosen to be smaller than the characteristic viscoelastic relaxation time (of order $10s$). We evolve the simulation for a total of n=20, 000 steps (4, 400s in real units). The equations of motion are solved using the Euler method for each vertex. If at any point during the simulation, the left and right vertex positions approach one another to a distance less than $0.5\mu m$, we label the junction as having successfully completed the shortening. The initial junction length was set to be $L_0 = 2$, equal to $20\mu m$ in real units. The range of rest length exponents we consider is limited by the need to ensure that the minima of the potentials do not overlap during a given simulation run. To generate the phase diagram for the probability of junction shortening as a function of the rest length exponent and the viscoelastic ratio, we consider for the left active vertex $0.05 \leq \frac{1}{\tau_L} \leq 5$ at intervals of 0.5. Rest length exponents in the range, $1.7 < \psi_L < 2$, were simulated at intervals of 0.25 for the active vertex. $\psi_R = 1.3$ is fixed for the passive right vertex. The acceleration of the potential minima, is set to be $\overline{a}_L = \overline{a}_R = 0.001$. The viscoelastic ratio for the right passive vertex is fixed at $\frac{1}{\tau_R} = 0.1$. We simulated 100 junction shortening events at each value of the parameters $\frac{k_L}{\gamma_L}$ and $\psi_L$. By monitoring the percent of successful junction shortening events, we generate the phase diagram (*Figure 2C*, Main Text).

## Section 8. Effect of viscoelasticity on the shortening of junctions

We observe in the phase diagram (*Figure 2C*, Main Text) that at a fixed value of the rest length exponent, modulating the asymmetry of the viscoelastic parameter $((\frac{k_L}{\gamma_L})/(\frac{k_R}{\gamma_R}))$ leads to a transition from non-shortening (failure to shorten) to junction shortening (successful shortening) regime. At constant $\psi_L = 1.95$, for low values of the active vertex viscoelastic parameter $(\overline{\tau}_R/\overline{\tau}_L) < 6.9$, less than 40% of the junctions shorten. However, at higher values of the viscoelastic parameter, $\overline{\tau}_R/\overline{\tau}_L > 15$, more than 80% of the junctions successfully execute shortening. Therefore, the theory predicts that the asymmetry in local viscoelasticity is critical for cells to intercalate medially and effect convergent extension. We calculate the normalized length for non-shortening junctions and found that the self-similarity in junction length dynamics is broken, in agreement with experimental results *Figure (2H,Main Text)*. $L_n$ for non-shortening junctions is characterized by large fluctuations away from the expected compressed exponential behavior, as quantified by the residue (see details below). Simulated junction length dynamics for the non-shortening case(gray curves in *Figure 2H* Main Text) is obtained for parameter values $1/\overline{\tau}_L = 0.05$, $1/\overline{\tau}_R = 0.05$ and $\psi_L = 2, \psi_R = 1.3$. Meanwhile, for the shortening phase (gray curves in *Figure 2G* Main Text), $1/\overline{\tau}_L = 5$,

$1/\bar{\tau}_R = 0.05$ and $\psi_L = 2, \psi_R = 1.3$. Therefore, asymmetry in viscoelasticity is critical for junctions to execute shortening. Our model points out that the persistent dynamics of active vertices, enabling the efficient shortening of the cell-cell interfaces, is a direct consequence of the faster viscoelastic relaxation time.

## Section 9. Normalized junction length dynamics

We calculated the normalized cell-cell junction contact lengths to characterize the self-similarity in the length change underlying cell neighbor exchanges during convergent extension. We selected all cell-cell contacts that shorten over time intervals > 100 s, and normalized the change in length as,

$$L_n(t) = \frac{L(t) - L(t_f)}{L(t_0) - L(t_f)} \tag{18}$$

where $L(t_f)$, $L(t_0)$ are the junction lengths at the final and initial time points respectively. The normalized junction length dynamics, $L_n(t)$, provides insight into the active processes that underlie vertex movement driving CE. Since junction lengths are highly heterogeneous (*Figure 2—figure supplement 1A*) relative to, $L(t_0)$, and the time to closure, $t_f - t_0$, the normalization in *Equation 18* allows us to rescale all the length changes to values between 1 and 0. The normalized length curve was smoothed (over 10-time frame windows = 20s) to remove higher frequency noise. To determine if junction shortening exhibits a self-similar behavior across multiple embryos, we rescaled the time axis in $L_n(t)$ by the relaxation time $\tau_f$, defined as the time at which $L_n(t = \tau_f) = 0.3$. This corresponds to a 70% reduction in the junction length. Rescaling the time axis by $t/\tau_f$ collapses the normalized lengths onto the functional form,

$$L_n[t/\tau_f] = e^{-1.5\left(\frac{t}{\tau_f}\right)^{3.8}} \tag{19}$$

which is a single compressed exponential (*Figure 2—figure supplement 1B*). The extent of the self-similarity is striking in comparison to both non-shortening (*Figure 2H*, Main Text) and cis-mutant normalized junction lengths (*Figure 6D*, Main Text). Notice that for $t < \tau_f$, change in normalized junction length is slower than exponential decay. However, for $t > \tau_f$, normalized junction length shortens significantly faster than would be predicted based on exponential decay. Therefore, the compressed exponential behavior for $L_n$ provides evidence that the persistence of junction shortening increases with time.

## Section 10. Residue

We quantify the deviation of the normalized junction shortening from the expected compressed exponential behavior by calculating the rescaled time, $t_r = \frac{t}{\tau_f}$, and $\omega = |e^{-1.5(t_r)^{3.8}} - L_n|$, where $\omega$ is the residue. $\tau_f$, is defined as the time at which $L_n(t = \tau_f) = 0.3$. In (*Figure 2I*, Main Text), non-shortening junctions show strong deviations from the expected compressed exponential behavior while shortening junctions closely follow the compressed exponential form.

## Section 11. Alternative form of the contractile force

To test the robustness of the conclusions obtained using our model, we consider an alternative form of the contractile force experienced by the vertices. We model the actuators contributing to viscous junction shortening as moving with constant velocities - $\bar{v}_L$ and $\bar{v}_R$- for the left and right vertices respectively:

$$\frac{d\bar{x}_L}{d\bar{t}} = -\frac{1}{\bar{\tau}_L} \times \left(\bar{x}_L - \bar{v}_L \bar{t}\right) + \bar{\zeta}_L \tag{20}$$

$$\frac{d\bar{x}_R}{d\bar{t}} = -\frac{1}{\bar{\tau}_R} \times \left(\bar{x}_R - \left(\bar{L}_0 - \bar{v}_R \bar{t}\right)\right) + \bar{\zeta}_R \tag{21}$$

The 'trap' potential in this scenario is of the form, $V_L = k_L(x_L - v_L t)^2$ and $V_R = k_R(x_R - (L_0 - v_R t))^2$, moving with constant velocities. Left vertex is defined to be active with $\bar{v}_L > \bar{v}_R$. The velocity is normalized as $\bar{v}_L = v_L / \left(\frac{x_0}{\tau}\right)$. The passive vertex velocity is fixed at $\bar{v}_R = 0.011$, which in dimensional units correspond to $0.0011 \mu m/s$. The active vertex velocity is varied in the range of $0.03 \leq \bar{v}_L \leq 0.034$, which in dimensional units is between $0.003 \mu m/s$ - $0.0034 \mu m/s$. Experimental vertex shortening velocities in the range of $0.001 \mu m/s$ to $0.021 \mu m/s$ was reported by some of us in a previous work (*Shindo and Wallingford, 2014*). Fixing the passive viscoelastic ratio at, $1/\bar{\tau}_R = 0.1$, we varied $0.05 \leq 1/\bar{\tau}_L \leq 5$ for the active vertex. Keeping all the other parameters the same, we arrive at the same conclusion that local junction viscoelastic response is critical to effect junction shortening (*Figure 2—figure supplement 1D*). Therefore, our conclusions are not affected by the specific form of the vertex dynamics. A crucial aspect is that the two potential well minima should move asymmetrically in time.

## Section 12. Transverse fluctuations of the vertices and Straightness Index

We quantify the intermittent movement (See *Figure 3A*, Main Text) of the vertices perpendicular to the motion that contributes to the junction shortening by calculating the transverse fluctuations, $R_T$. The transverse step size is given by, $\delta r_T(t) = \left|\delta \vec{r}_L(t)\right| sin(\theta)$, where $\delta \vec{r}_L(t) = \vec{r}_L(t) - \vec{r}_L(t - \delta t)$ and the angle $\theta$ is the obtained from the dot product, $\delta \vec{r}_T . \Delta \vec{r}_L = |\delta \vec{r}_T||\Delta \vec{r}_L| cos(\theta)$. Here, the net displacement of the Left(L) vertex is given by, $\Delta \vec{r}_L = \left(x_L(t_f) - x_L(t_0)\right)\hat{x} + \left(y_L(t_f) - y_L(t_0)\right)\hat{y}$. Similar equation applies for the right vertex with $x_L, y_L$ replaced by $x_R, y_R$. To better quantify the intermittent dynamics, we compute the transverse 'hop' function,

$$R_T(t) = \left(\delta r_T(t) - \langle \delta r_T \rangle_B\right)^2 \tag{22}$$

The angular bracket above $\langle .. \rangle_B$ denote the average over the time window $BNoentity[t - \delta t, t + \delta t]$. We chose for the hop duration parameter, $\delta t = 4s$, to probe short time transverse fluctuations. The probability distribution of all $R_T(t)$ values are shown in (*Figure 3D*, Main Text). By averaging the transverse fluctuations over all vertices, $\langle R_T(t) \rangle = \frac{1}{N}\sum_{i=1}^{N} R_T(t)_i$, we obtain the mean transverse fluctuation for active and passive vertices (*Figure 3C*, Main Text). $\langle R_T(t) \rangle$ for Cis-mutant and Xdd1 vertices are shown in *Figure 6F* and *Figure 7F* of the Main Text, respectively.

### Straightness index

The directionality of the vertex trajectories were assessed using the straightness index. This is defined as the ratio of the net distance moved by a vertex between initial and final time points to the total distance moved by a vertex:

$$Straightness\ Index = \frac{\left|\vec{r}(t_f) - \vec{r}(t_o)\right|}{\Sigma_t \left|\vec{r}(t + \delta t) - \vec{r}(t)\right|} \tag{23}$$

Higher the value of the straightness index, the more directed the movement is with the value of straightness index = 1 indicating perfectly straight line motion (see *Figure 3E*, Main Text).

## Section 13. Cadherin clustering from the spatial autocorrelation function

To determine the characteristic spatial correlation of cadherin intensity fluctuations, we analyze the pixel-by-pixel Cadherin3 (Cdh3) intensity data, $I\left(\vec{r}_i\right)$, along the medio-lateral cell-cell interface (v-junction). Here, $\vec{r}_i$ is the position of the i-th pixel in the iSIM image. The spatial autocorrelation function of the cadherin intensity fluctuations as a function of distance, r, along the cell-cell interface is,

$$C(r) = \sum_{i,j} \theta\left(r - \left|\vec{r}_i - \vec{r}_j\right|\right) \left[\frac{\left(I\left(\vec{r}_i\right) - \langle I \rangle\right)\left(I\left(\vec{r}_j\right) - \langle I \rangle\right)}{\langle I^2 \rangle - \langle I \rangle^2}\right], \tag{24}$$

where $\theta(z) = 1$ if $z = 0$, $\theta(z) = 0$ for any other value of $z$. $\langle I \rangle$ is the mean cadherin intensity over all the pixels along the cell-cell junction. $C(r)$ is normalized such that $C(r = 0) = 1$. The cadherin correlation length is defined as the distance, $\xi$, at which $C(r = \xi) = 0$. This provides a measure of the distance scale at which the correlation in cadherin intensity fluctuations is lost. Equivalently, $\xi$, sets the spatial persistence of cadherin fluctuations along the cell-cell junction, providing a quantitative measure of lateral cadherin clustering. We analyzed cadherin clustering patterns along individual cell-cell junctions separately at time intervals of 2s and obtained the spatial correlation behavior for individual junctions from 10 embryos. The mean of the cadherin spatial correlation (over 100s of time points) for wild type embryos is reported in *Figure 4C*, Main Text with the error bar denoting the standard deviation. To analyze the dynamic variation in cadherin cluster size as a function of time, $C(r)$, was calculated over a time interval of 320s at 2s resolution. The fluctuation in cluster size is given by, $\delta\xi(t) = \xi(t) - \langle\xi\rangle_t$, where $\langle\xi\rangle_t$ is the mean cluster size over the analyzed time interval (, 4E Main Text). The cluster size fluctuation, $\delta\xi(t)$, was smoothed (over 10-time frame windows = 20s) in order to remove high frequency noise.

## Section 14. Junction length fluctuations

To analyze the instantaneous change in the junction length, we calculated, $\delta L(t) = (L(t) - L(t + \delta t))$, where $\delta t = 2s$ and $t$ is the time. The unit of the length fluctuations is $\mu m$. When the junction shortens, $\delta L(t) > 0$, while extension implies $\delta L(t) < 0$ (*Figure 4E*, Main Text).

## Section 15. Cross-correlation between cadherin cluster size and the junction length fluctuations

The normalized cross-correlation between junction length fluctuations, $\delta L(t)$, and cadherin cluster size fluctuations, $\delta\xi(t)$, was calculated in MATLAB using,

$$C_{\delta L, \delta\xi}(\tau) = \int_0^T \delta\xi(t) \times \delta L(t + \tau)dt, \tag{25}$$

where T is the total time of analysis and $\tau$ is the lagtime. We analyzed the cross-correlation for 18 junction shortening events and show the correlation coefficient as a heatmap in *Figure 4F*, Main Text

## Section 16. Asymmetry in cadherin clustering:

To quantify the asymmetry in Cdh3 clustering in the spatial region near the left and right vertices, we calculated the spatial correlation in cadherin intensity fluctuations, $C(r)$ (see *Equation 24*), in a region spanning $3.25\mu m$ adjacent to left and right vertices. The spatial region is chosen such that on average it is $3X$ larger than typical cadherin cluster size of order $1\mu m$. The localized cadherin clustering behavior adjacent to active and passive vertices, quantified by the spatial correlation in cadherin intensity fluctuations, is shown in *Figure 4G* Main Text. Main Text also compares the local cadherin clustering behavior in non-shortening junctions to shortening junctions.

We then wondered if spatial correlation in cadherin expression is stable in time across a single cell-cell junction. We computed the spatial cadherin correlation at 2 second time intervals along single cell-cell junctions from multiple embryos. Our results indicate that the spatial correlation length of cadherin expression is highly heterogeneous in time, with the correlation length varying from 0.2 $\mu m$ to 1.6 $\mu m$, as shown in *Figure 4—figure supplement 2*. To decipher how the spatial correlation in cadherin expression along the cell-cell junction varies with time near active and passive vertices, we present the data for spatial autocorrelation of cadherin fluctuations in *Figure 4—figure supplement 2A-J*. In each panel, individual blue (red) lines correspond to the spatial correlation in cadherin fluctuations near passive (active) vertices. The difference in spatial cadherin correlation between active and passive vertices in shortening junctions are shown in *Figure 4—figure supplement 2A-J*

whereas the same data for non-shortening junctions are shown in *Figure 4—figure supplement 2K-Q*. By extracting the length of the spatial correlation as discussed in Section 6 above, the summary of the cadherin spatial correlation data is presented in the Main Text *Figure 4G*.

We used an alternative definition of C-cadherin cluster size to confirm our results. By fitting the decay in the cadherin spatial autocorrelation function to zero by an exponential function, we can extract the cluster size. We find that the asymmetry in the local cadherin clustering behavior is independent of the definition of the cluster size (*Figure 4—figure supplement 1E–G*). Hence, we conclude that cadherin clustering is enhanced near active vertices as opposed to passive vertices in shortening junctions (*Figure 4G* Main Text, *Figure 4—figure supplement 1F*) while it is symmetric near left and right vertices in non-shortening junctions ( *Figure 4—figure supplement 1G*).

## Section 17. Perturbation of Cadherin clustering in individual junctions and its spatial periodicity

We calculate the spatial autocorrelation of the cadherin intensity fluctuations ($C(r)$ **see Section 13,** *Equation (24)*) for four different embryo development scenarios, (i) wild type Cdh3 *Figure 4C*,(ii) Cdh3 rescue (Cdh3-GFP) (see *Figure 7—figure supplement 1*), (iii) Cdh3 cis-mutant (cisMut-Cdh3-GFP) *Figure (5D,Main Text)*, and (iv) Xdd1 *Figure (7C,Main Text)*. The mean spatial correlation in cadherin fluctuations for wild type Cdh3 and Cdh3-GFP junctions show similar behavior with the decay to zero characterized by an exponential form (*Figure 7—figure supplement 1B*). The exponential spatial dependence is evidence for the existence of a characteristic spatial scale for correlations in cadherin spatial distribution. The local peak in the cadherin autocorrelation function is identified using the *findpeaks* algorithm in MATLAB. Local peak in a data array is identified when a data point is larger than its two neighboring data points or equal to infinity. The prominence of the peak is set to 0.1, identifying the peak amplitudes that stands out relative to other peaks. Wild type and rescue embryo cell-cell junctions are characterized by well-defined spatial periodicity in cadherin clustering, as observed from the secondary peaks in the spatial correlation (see black triangles, *Figure 7—figure supplement 1C*). Therefore, cadherin spatial organization in wild-type Cdh3 and Cdh3-cis-mutant rescue embryos is in a crystal-like phase ( *Figure 7—figure supplement 1C*), with regularly repeating spatial patterning. Cadherin spatial correlation for individual frames (with no averaging) is shown for control (solid lines in ; *Figure 7—figure supplement 1D*) and cadherin rescue embryos (dashed lines, *Figure 7—figure supplement 1D*). However, cadherin spatial correlation in Xdd1 and cisMut-Cdh3 embryos show diffuse spatial organization, with little to no secondary peak structures visible in the spatial autocorrelation (see *Figure 7—figure supplement 1E*). This is indicative of disrupted periodicity in cadherin clustering. Therefore, cadherin spatial organization in cisMut-Cdh3 and Xdd1 embryos is in a gas-like phase. Individual frame cadherin spatial correlation for cisMut-Cdh3 (solid lines *Figure 7—figure supplement 1F*) and Xdd1 is shown as dashed lines in *Figure 7—figure supplement 1F*. For Xdd1 and cisMut-Cdh3-GFP junctions, the decay in the spatial correlation is better fit by a power law, indicating the lack of existence of a coherent length scale associated with fluctuations in cadherin expression ( *Figure 7—figure supplement 1G*).

## Section 18. Statistics

The statistical test used and other relevant details such as the number of embryos/image frames analyzed are described in the figure legends.

