## [Decision Letter]

**Acceptance summary:**

This study provides a conceptually novel perspective on the mechanics of cell-cell junctions, and how they are regulated for morphogenetic cellular rearrangements. The cell biology of junction dynamics is a very topical subject and should be of broad interest to cell and developmental biologists interested in the mechanics of collective cell motion. It provides new biological insights combined with new theory for junctional mechanics, and is a great example of synergy between biological experiment and physics-based theory.

**Decision letter after peer review:**

Thank you for submitting your article "Lateral clustering of cadherins imparts mechanical heterogeneity to single cell-cell junctions during axis elongation" for consideration by *eLife*. Your article has been reviewed by 3 peer reviewers, and the evaluation has been overseen by a Reviewing Editor and Kathryn Cheah as the Senior Editor. The following individual involved in review of your submission has agreed to reveal their identity: Α Yap (Reviewer #3).

Summary:

The authors present a detailed kinematic analysis of vertex dynamics and evaluate the role of cis-cadherin interactions in bicellular junctions during cell intercalation of *Xenopus* gastrulation. This is a very topical subject as there have been several papers in the past three to four years on the cell biology of junction dynamics in cell culture systems and in *Drosophila* models of epithelial morphogenesis. The paper provides new biological insights combined with new theory for junctional mechanics. It is a great example of synergy between biological experiment and physics-based theory.

Essential revisions:

1. The authors propose that motions of two vertices connecting shrinking junctions are independent; one vertex moves while the other remains fixed in place. It is essential that this is not a point-of reference artifact, and there are concerns about the methodology used for this analysis as the descriptions are unclear. The diagram in the supplement shows that 1 landmark is used to measure the position of the two ends of the v-junction; but a second independent point is needed for true triangulation. The second issue is that the relative positions are not measured against a fixed spatial frame but rather against local structures whose movements may be coupled to movement of the tricellular junction of interest. More details on the methodology need to be provided, and the analysis may need to be redone if indeed only 1 landmark was used.

2. A more extensive characterization of the cis-Cdh3 mutant is needed to interpret its effect on cell intercalation and vertex motion. Specifically, does the cis-Cdh3 mutant have non-junctional effects on actomyosin and cell protrusivity in a cell-autonomous manner? A cell-autonomous, non-junctional alteration in actomyosin could also lead to the observed effects on junction shrinkage and convergent extension. These data should be incorporated into a more formal hypothesis/model for the role of cis-Cdh3 interactions in vertex motion.

3. An alternative interpretation for presence of an active vertex is that it reflects the leading edge of a cell protruding mediolaterally between adjacent cells. Interestingly, another manuscript from the Wallingford lab has been posted onto BioRxiv (Weng, Huebner, and Wallingford, 2021) that includes a description of F-actin rich protrusions from the mediolateral neighbor. This neighbor protrudes, displacing the tricellular junction, and the anterior-posterior bicellular junction shrinks. Thus, the "stiff" tricellular junction, is shared by a cell with an actin-rich protrusion. This may fully explain the distinct microrheological character of the moving vertex. The authors should comment on this observation and provide a rationale for why they are not testing this hypothesis.

4. Several aspects of the methodology are missing. For instance, what criteria are used to distinguish shortening from non-shortening junctions? How are 'active' and 'passive' vertices defined? When the authors introduce their Straightness Index, it is not clear from the Results that they are talking about the straightness of motion of the vertices. There are few details about the imaging conditions, such as the z depth of the images and the time interval between frames.

5. A broadened discussion of how the findings fit (or not) into the current model of convergent extension and roles of PCP in this process is needed. Is there any known connection between the PCP machinery and Cdh3 cis-clusters? How can PCP contribute to this mechanical heterogeneity along the mediolateral axis? Additionally, further discussion of where the directionality for contraction derives from is needed. Do clustered cadherins impose a directionality to the active vertex movements?

*Reviewer #1:*

In follow-up to previous studies from the Wallingford group, the focus here is on junction dynamics in the anterior and posterior junctions of actively intercalating prospective notochord cells. The advances in characterizing vertex dynamics emerge from incorporating concepts from soft matter physics and microrheology. The conceptual framework borrowed from physics is an interesting move in the analysis of cell boundary motions in this classical system of directed cell rearrangement but I am not fully convinced that the framework brings new understanding to this problem. The joining of the conceptual framework of vertex analysis and the bicellular study of cis-cadherin would seek to bend principles borrowed from apical junction dynamics and apply to mesenchymal junction dynamics but it has not been made sufficiently clear what new insights this brings.

1. What exactly is "mechanical heterogeneity" and what precise role does it play in cell intercalation and convergent extension? Is it necessary, sufficient, or merely coincident with the process? Answering these questions is critical to the potential impact of this paper on the field. There are many heterogeneous events and processes throughout a single embryonic event but why is this one so important?

2. Very much like the observations from Vanderleest et al. (2018, *eLife*), the authors propose that motions of two vertices connecting shrinking junctions are independent; one vertex moves while the other remains fixed in place. This seems plausible but I have two issues with their analysis. The first is that the authors claim to use triangulation to track junctions in the "lab frame of reference", but I understand that triangulation requires 2 landmarks to track the absolute position of 1 point. The diagram in the supplement shows that 1 landmark is used to measure the position of the two ends of the v-junction; a second independent point is needed for true triangulation. The second issue is that the relative positions are not measured against a fixed spatial frame but rather against local structures whose movements may be coupled to movement of the tricellular junction of interest. Motions of yolk platelets or other tricellular junctions from one of the 3 cells surrounding a specific tricellular junction are likely to be mechanically coupled to that same junction. It would be prudent to use a platelet or junction from cells that are not one of these 3 cells.

3. This reviewer enjoyed the descriptive focus on vertex kinematics from the microrheological perspective, however, some background on the pro's and con's of the methodology (e.g. Crocker and Hoffman, Meth Cell Bio, 2007) would be helpful. Furthermore, the level of processing to quantify vertex fluctuations passed over several key features of vertex motion that would be of great interest to the community and should be included. For instance, it would be very helpful to describe advective kinematics of these structures starting with displacement vs. time kymographs, and extending to velocity and persistence. Are there any correlations between the active vertex and lateral (or medial) vertex motions? I understand that these vertices move in a non-uniform manner but would appreciate more intuitive description of their motion as if they were, for instance, a set of cell protrusions. A detailed "protrusion-like" analysis has not been carried out for these structures and would be useful to compare to lamellipodial kinematics and leading edge fluctuations described at the substrate level of intercalating cells.

4. This analysis suggests a novel function for tricellular junctions in mesenchymal tissues. In contrast to the tricellular junction of an epithelial sheet, this structure is not point-like but would need to be considered as a tricellular boundary that extends from the superficial, extracellular matrix facing cell surface to the most dorsal face of the notochord cell. Rather than a point, this junction is more like a knife-edge and can exhibit a complex topology typical of "escutoid cells" described by the Escudero lab (Nat Comms, 2018) or the Toyama lab (Nat Cell Bio, 2017). This raises questions about the position of the vertices measured and whether the kinematics are heterogeneous along all points of the junction, are they consistently active, or are some points advancing while others are retracting. Have the authors described kinematics of the same vertex at different z-positions?

5. Several aspects of the methodology are missing. For instance, what criteria are used to distinguish shortening from non-shortening junctions? There are few details about the imaging conditions, such as the z depth of the images and the time interval between frames.

6. The authors stress that their model is unique in driving cell intercalation but there are several theoretical and computational models have previously demonstrated the role of either mediolateral cell protrusions or anisotropic contractile bicellular junctions in directing mediolateral cell intercalation. Multiple examples of these models including recent one by Belmonte and co-workers (PLoS Comp Bio, 2016) involve heterogeneous vertices, while others not requiring heterogeneous junctions, are sufficient in driving convergent extension. The compressed exponential fits the spatial and temporal scaled kinematic changes of the shrinking bicellular junction but it is not clear whether this is simply a mechanistically-based phenomenological principle or merely a "good-fit" to the data.

7. An alternative interpretation for presence of an active vertex is that it reflects the leading edge of a cell protruding mediolaterally between adjacent cells. Why is this possibility not discussed?

8. The second half of the manuscript describes an important role for cis-cdh3 interactions in establishing or maintaining cell behaviors and polarity within converging and extending cells. The phenotypes are clearly demonstrated at the embryonic and tissue level. The subcellular phenotypes are also well described but it is not clear that the kinematic phenotypes are directly responsible for the failure of these cells to undergo directed cell rearrangement. Additionally, the cell autonomy of the defect is not clear. The authors would need to show that cis-interactions in anterior and posteriorly apposed cells alone are responsible for defects in rearrangement. For instance, would a wild-type lateral-cell, or one-lateral and one-AP cell be able to rescue polarity and active kinematics of the vertex?

9. Given the role of the actin cytoskeleton in directed cell intercalation I feel the role of cis-cdh3 interactions is only half-explored. What changes in protrusive activity, and actomyosin dynamics are perturbed? Are these specific to PCP polarity machinery operating in the mesoderm or can they also be observed in cis-cdh3 deficient non-polarized cells such as non-neural ectoderm cells (see Kim, J Cell Sci, 2011)? Up and down regulation of PCP signaling similarly regulates actomyosin contractility in both mesoderm cells and non-neural ectoderm – so this should be straightforward to test.

10. Overall, more discussion is needed about how these kinematics would fit or not fit into the current model of convergent extension and roles of PCP in this process. Is there any known connection between the PCP machinery and Cdh3 cis-clusters? How can PCP contribute to this mechanical heterogeneity along the mediolateral axis?

*Reviewer #2:*

The authors suggest that tricellular vertices located at opposite ends of cellular interfaces can show independent displacements, similar to what has been reported during *Drosophila* convergent extension movements. This is an important finding, although there are concerns about how well the analysis validates this point, as well as the brevity at which methods are explained. The study shows that these vertices demonstrate either active (directed) or passive (diffusive) movements, which, in turn, suggests that there must exist local forces that drive these specific vertex behaviors. The authors then explore what changes in the predicted cellular viscoelastic properties could be sufficient to cause vertex displacements through a mechanobiological modeling approach – these results suggest that local stiffening and relaxation of interfacial forces may be responsible for the vertex movements. To test their modeling predictions, the authors examine the distribution of cadherin proteins and observe that cadherins often cluster along cell interfaces, and that this clustering is asymmetrical at vertices that are showing "active" regimes of displacement. Finally, the authors demonstrate the cadherin clustering is regulated by PCP signaling.

There are several potentially nice findings from this study. One weakness is that the methods (especially the computational methods) are not well-described, and the findings are not detailed well in the manuscript. It is often stated that a finding supports a particular hypothesis, followed by a reference to a figure, but there is not a detailed description of what the findings actually are and how this specifically supports the hypothesis. At times, this makes the manuscript read superficially. I believe the Results section needs substantial re-writing to more adequately explain the computational sections (~first half of manuscript). The Discussion is also brief in content. More specific comments follow below:

1. I thought the authors had a bit of an odd way of introducing the work, especially the paragraph suggesting "a similarly granular understanding of subcellular mechanical properties" has not been achieved. (From Intro, "For example, the localization and turnover of actomyosin and cadherin adhesion proteins have been extensively quantified during *Drosophila* CE (Blankenship et al., 2006; Fernandez-Gonzalez et al., 2009; Levayer and Lecuit, 2013; Rauzi et al., 2008)), as have similar patterns for the Planar Cell Polarity (PCP) proteins and actomyosin during vertebrate CE (Butler and Wallingford, 2018; Kim and Davidson, 2011; Shindo and Wallingford, 2014). However, the significance of these molecular patterns remains unclear because we lack a similarly granular understanding of subcellular mechanical properties and their dynamics, which ultimately explain the cell behaviors that drive CE.".

I think the authors have a potentially interesting study, but it does not appear to address a significantly different scale than what many studies out of the Lecuit lab (or others) have addressed (e.g., Fernandez-Gonzalez, Gardel, Campos, Dahmann, Kiehart, or Hutson labs). It would be appropriate to better acknowledge prior work – it does not detract from the accomplishments of this study. My apologies, but at times the writing, especially the physical overlay of the paper, sounds dismissive of other works.

2. It is not always clear how some of the biophysical analysis are performed and/or support the authors major contentions. One example, "We found that v-junction shortening was dominated by the movement of a single "active" vertex, while the other "passive" vertex moved comparatively less (Figure 1C, D)(SI, Section 1). Three distinct metrics demonstrated that this asymmetry was not a point-of-reference artifact (Supp. Figure 2)."

While I suspect they are correct, this is one major sticking point and it appears the authors are not meeting scientific standard in demonstrating this point in its current form. First, the wording 'Three distinct metrics demonstrated that this asymmetry was not a point-of-reference artifact' seems to suggest that the authors are showing three independent pieces of evidence, but this is not the case (maybe this impression was not intended, so maybe reword?). What the authors are actually showing are the MSD results in three different coordinate systems (the 'lab' frame, the 'cell' frame which is a reference frame relative to a yolk particle, and the reference frame relative to a non-moving neighboring vertex). As best as I can guess, this is the comparison of the results from the three reference frames that would (potentially) allow the conclusion that the different motion in the two vertices is not an artifact, but this is not possible to tell, because the authors only describe their results in the supplementary figure, and do not explain anywhere how/why these three different results prove the point that the different motion cannot be a reference point artifact. The mere fact that the authors were able to identify three different reference frames in which the vertices move differently does not meet this threshold; they would have to explain better why one of them (e.g. the cell frame) represents something like a fixed local coordinate system, and – for example – show that an ensemble of yolk particles doesn't 'deform' significantly over the associated time scales (for example, they could show that the yolk particles only minimally changes their interparticle distances during this time?).

Additionally, the authors write in the figure legend 'We refer to this as the relative MSD in the "cell" frame of reference, which eliminates the effect of translational and rotational motion of the tissue during time-lapse imaging.' What the authors show in the figure is a single yolk particle as reference point, and a single reference point would only be able to eliminate the effect of translation, but not translation and rotation at the same time. So are the authors using only a single yolk particle for a set of two vertices and ignoring rotation effects? Or are they using a system where they can construct a coordinate system from multiple yolk particles simultaneously (which could correct for both translation and rotation) but not showing this procedure in the figure? And going forward, which reference frame are the authors using for their analyses, since all three show different results, and why? These things would be best explained explicitly in a section of the supplement.

3. Where does the directionality for contraction derive from? I found the authors "transverse" vs "in-line" coordinate system somewhat confusing. Transverse in other studies has often been applied to the non-contracting interface, but lines drawn in figure schematics seem to indicate a more cell-center axis. I believe the authors are showing more clustered, stable adhesions correlating with "active" vertices, but does this impose a directionality to the active vertex movements (in the experimental data sets)? Any further description would be helpful, and this could also be expanded on in the Discussion.

4. Some of the MSD difference between active and passive vertices appear small. It is not clear how we know that the differences between active and passive vertices represent something 'real' and not an artifact of the classification process. In the first section of the results, the authors are comparing various metrics (such as MSD) between 'active' and 'passive' vertices, but I am not finding a single clear explanation of how these categories were defined. Are the authors using a threshold to classify a given vertex as active or passive? Or are the authors taking vertex pairs, and splitting up each pair into one active and one passive partner, based on MSD magnitude? This is a critical piece of information for interpreting the results. Can the authors show that there are two 'modes' of behavior (active and passive) using a clustering approach of some type?

5. I would like to hear more interpretation of what the mechanical properties of the cells are that are being affected in the -cis binding mutants (this is taken somewhat as canon, but does it behave the same in these *Xenopus* explants?). I know the authors say cell-cell adhesion is (somewhat?) rescued by this -cis mutant and embryos do not fully dissociate. Can the authors better detail the degree to which the strength of cell-cell adhesion is rescued versus local properties like the asymmetry of adhesive forces at cell vertices?

6. A final note about the modeling. About their model predictions, the authors state that 'Our theory makes a curiously counter-intuitive prediction: that the more fluid-like motion of the active vertex occurs in the context of increased local stiffness (i.e. higher viscoelastic parameter), while the more glass-like motion of the passive vertex occurs in a relatively decreased stiffness regime.'

I am not sure that there is something counter-intuitive about this finding. While it is true that in the classic model for epithelial morphology (Farhadifar et al. 2007), the more fluid-like behavior is associated with lower stiffness, that's because it's related to adhesion forces dominating over elastic forces, so the asymmetric vertex movement is being driven by the minimization of adhesion energies. In the authors' model, there are no adhesion forces, and the vertex displacement is driven by the external piston forces – so it seems logical that the more effective motion (in the active vertex) would occur in the case where the higher spring stiffness allows better force transmission (and the passive vertex is the one where the forces dissipates more quickly in the softer environment). Thus, the fluidization arguments from other models/systems in the literature would not seem to fully apply to this system.

7. "Symmetrical clustering in non-shortening reflects the symmetrical dynamics of vertices bounding these junctions" – I believe "reflects" suggests a direct connection, whereas this is more of a "correlates" or "suggestive" statement.

8. Figures often have panel titles like "Deviation from compressed exponential" – it would helpful to better title these to indicate the rough property that the authors are trying to report on, and put more specificity into the figure legends.

*Reviewer #3:*

I think that this is an interesting and important report, for the following reasons. (1) It introduces the concept that cell-cell adherens junctions are mechanically heterogeneous and this mechanical asymmetry mediates the changes in junctional length that underlie convergent extension. And it introduces interesting new theory that incorporates this concept. (2) The authors provide a biological function for cadherin clustering, something which has not been available to date. Importantly, their explanation bridges scales, i.e. they show that clustering (a molecular event) contributes to convergence extension by modulating junctional mechanics (a cellular biophysical parameter). In general, the data are of excellent quality and the study represents an exemplary case of synergy between experiment and theory.

I suspect that this study is not the end of the matter. In particular, it would be important to understand how local differences in cadherins affect junctional mechanics. The authors cite earlier work from the Mege lab indicating that cadherins can influence membrane mechanics. Whether this is at the level of the bilayer or via alterations in the cortical cytoskeleton would be important to address. But this is beyond the reasonable scope of a paper; indeed, it would speak to the potential for this study to motivate future research.

1. Figure 4G. Is this correct as currently presented? The authors describe this as showing that cadherin cluster size near non-shortening vertices is the same as that around the passive vertices in shortening junctions. But to my eye, what is drawn as the non-shortening data set looks similar to the set that is labelled as (active). Has something been mislabelled?

2. Embryo cohesion. This is measured as embryo survival at stage 23. But that seems to me to be a very indirect measure of tissue integrity. Changes in survival may reflect catastrophic loss of integrity, but miss more subtle features. I would be happy enough if they were to use the term "viability" instead of "cohesion". There is still a very interesting different in effect of the cis-mutant.

3. I think that some of the analyses could be explained more intuitively for the general reader. For example, when the authors introduce their Straightness Index, it is not clear from the Results that they are talking about the straightness of motion of the vertices. (For junction folks, "straightness" sounds like the measures of junctional straightness that have been used as proxies of line tension.) It would only take a sentence to give an intuitive explanation in this and other instances.

(By the way, the text and caption to Figure 3 give the reference for the Straightness Index as SI Section 14, but I think that it is actually Section 16.)

4. Glass-like behaviour in junctions. To what extent can one really say that the passive junctions are "more glass-like". If I read Figure 1E correctly, the exponent of "t" for the passive junctions is still >1 (1.3 from Figure 1E). So, is it more helpful to simply regard them as less fluidized, rather than invoking subdiffusive behaviour (which would require the exponent to be <1)? I'm happy to the corrected by the authors (or the other reviewers), but beyond a point of phenomenology I'm still not sure that I can see an underlying intuition that would be really helpful.

Along these lines, is it possible to statistically compare the differences in the exponentials?

5. "Non-shortening" junctions. Were these tested away from junctions that shortened? What about the junctions that also contribute to the vertices at the ends of the junctions that shortened? Did these show any asymmetry in mechanics or in cadherin clustering?

---

## [Author Response]

Essential revisions:1. The authors propose that motions of two vertices connecting shrinking junctions are independent; one vertex moves while the other remains fixed in place. It is essential that this is not a point-of reference artifact, and there are concerns about the methodology used for this analysis as the descriptions are unclear. The diagram in the supplement shows that 1 landmark is used to measure the position of the two ends of the v-junction; but a second independent point is needed for true triangulation. The second issue is that the relative positions are not measured against a fixed spatial frame but rather against local structures whose movements may be coupled to movement of the tricellular junction of interest. More details on the methodology need to be provided, and the analysis may need to be redone if indeed only 1 landmark was used.

We agree that it is essential to show vertex movements are not a point-of-reference artifact. We regret, therefore, that the original manuscript was not sufficiently clear on this point. In fact, what we call the “lab frame of reference” provides the “fixed spatial frame” the reviewers were concerned was missing.

As now clarified with a new schematic in the revised Figure 1—figure supplement 2, the “lab frame of reference” is a fixed coordinate system, where we used the pixels of the image to establish the coordinates with the upper left pixel set at (x,y)(0,0). This fixed lab frame of reference was used to generate the data in Figure1 and in Figure 1—figure supplement 1. These new points are discussed on page 4 of the revision.

Accordingly, the measures made using internal landmarks (yolk platelet, neighboring vertex) are *secondary* to our measurements using the fixed coordinate system. Nonetheless, we feel these are important to address the potential artifacts that might be introduced into our fixed frame quantification by overall tissue drift in the microscope. Thus, we further strengthened these *secondary* analyses by adding MSD plots for the yolk particles and the nearby vertices used as landmarks in Figure 1—figure supplement 2 (see panels E and G) showing that these landmarks move less compared to the vertices. We clarified this issue on page 4 of the manuscript.

Finally, to avoid confusion, we have removed the word “triangulation” from the figure legend.

2. A more extensive characterization of the cis-Cdh3 mutant is needed to interpret its effect on cell intercalation and vertex motion. Specifically, does the cis-Cdh3 mutant have non-junctional effects on actomyosin and cell protrusivity in a cell-autonomous manner? A cell-autonomous, non-junctional alteration in actomyosin could also lead to the observed effects on junction shrinkage and convergent extension. These data should be incorporated into a more formal hypothesis/model for the role of cis-Cdh3 interactions in vertex motion.

A more thorough characterization of the cis-Cdh3 mutant will undoubtably provide exciting results, but we respectfully argue that such work is beyond the scope of the present paper, for reasons both philosophical and practical:

Philosophically, we would argue that a deeper dive into Cadherin function will not enhance the results of *this* manuscript. Our results that Cdh3 clustering is required for an essential developmental process represent a significant step, as this has not been shown previously in the context of any intact animal. Moreover, we link clustering to the normal execution of novel mechanical behaviors that we also report here for the first time, providing a mechanistic link between subcellular mechanics, cadherin clustering and an embryonic phenotype. Finally, we link these biological phenotypes to fundamental physical attributes that we also report here for the first time. Thus, we feel that further dissection of any single aspect of this paper (e.g. clustering) would actually distract the reader from what is already a lengthy story arc. We hope the reviewers and the Editor can be convinced to agree!

From a practical standpoint, we note as well that testing cell autonomy of our cisMut-Cdh3 reagents in vivo is perhaps less straightforward than it seems: The experiment would require examining mosaics, and to generate these, we would inject single blastomeres at late cleavage stages and rely on CE-mediated cell mixing to place control cells next to experimental cells. However, since cisMUT-Cdh3 blocks intercalation, no cell mixing will occur, precluding accurate assessment of cell autonomy. At the very best, we may be able to image cell behaviors at the edge of a clone, a half-measure by any measure. We are currently exploring optogenetic approaches by which we may explore this important issue, but we feel those experiments are far beyond the scope of the present work.

3. An alternative interpretation for presence of an active vertex is that it reflects the leading edge of a cell protruding mediolaterally between adjacent cells. Interestingly, another manuscript from the Wallingford lab has been posted onto BioRxiv (Weng, Huebner, and Wallingford, 2021) that includes a description of F-actin rich protrusions from the mediolateral neighbor. This neighbor protrudes, displacing the tricellular junction, and the anterior-posterior bicellular junction shrinks. Thus, the "stiff" tricellular junction, is shared by a cell with an actin-rich protrusion. This may fully explain the distinct microrheological character of the moving vertex. The authors should comment on this observation and provide a rationale for why they are not testing this hypothesis.

This is a very important point, and we regret not addressing it in the original manuscript. In truth, the paper was so lengthy already that we sought to simplify it by not discussing actin. We see now that this was a mistake. We therefore have now added data that address this issue.

We quantified actin intensity in the regions of shortening v-junctions near active and passive vertices. As we now show in Figure 4—figure supplement 1, we observed no asymmetry in this region (which is the same region in which we observed asymmetry of cadherin clustering). These data, presented on page 7 of the revision, argue *against* the simple explanation that active and passive vertices are the result of asymmetric protrusive activity and cell crawling.

In addition, we note that the reviewers’ statement above concerning the data in our preprint from Weng et al. is in fact *not* accurate. The review states: “This neighbor protrudes, displacing the tricellular junction….” But this is not what we reported. Rather, those experiments were carefully designed to identify tricellular junctions using a membrane marker *independently* of the actin signal used to observe protrusions. This approach allowed us to observe that in *some* cases protrusion-related actin assembly is associated with vertex motion, but in an equal number of cases, it clearly is not. Thus, there is *not* a simple relationship between actin-rich protrusions and the active or passive vertex. We believe that a careful look at the data in Figure 2, Figure 1—figure supplement 2 and Supp. Figure 4 of the Weng et al. preprint should make this point clear.

Thus, while the reviewers are correct that understanding the relationship between actin and vertex motion will be critical, that relationship is quite complex. Thus, we hope that the new data in Figure 2—figure supplement 1 of the revised manuscript will suffice in the context of this paper.

4. Several aspects of the methodology are missing. For instance, what criteria are used to distinguish shortening from non-shortening junctions? How are 'active' and 'passive' vertices defined? When the authors introduce their Straightness Index, it is not clear from the Results that they are talking about the straightness of motion of the vertices. There are few details about the imaging conditions, such as the z depth of the images and the time interval between frames.

We wish to apologize to the reviewers here, as we realize now that much of the original supplemental information (appendix) was incorrectly organized, so that many relevant sections were not called out properly in the main manuscript. As a result, the manuscript appeared to be lacking methodological details.

To correct these mistakes, we have reorganized the Appendix, updated the Appendix callouts in the main manuscript, updated the experimental methods section, and added text to the main manuscript to clarify our methodology. Details for each specific point are below.

1. Criteria for distinguishing shortening vs non-shortening junctions. Any junction not shortening over a 400s timescale was considered non-shortening. We have added this criterion for junction shortening to the main text (pg.5) as well as a callout to the Appendix (section 1).

2. Definition of ‘active’ and ‘passive’ vertices. The quantitative definition of the active and passive vertex is also given in the Appendix (section 1). To clarify this point, we have added a brief description of the method used to define the active and passive vertex to the main manuscript (pg.4) as well as a callout to SI Section 1, as follows:

3. Definition of Straightness Index. Our Straightness Index is defined in detail in the Appendix Section 12 and refers to vertex motion and not bicellular junction “straightness”. To clarify this point, we added the following text to the main manuscript (pg.6), as well as a callout to SI Section 12.

4. Missing details of imaging conditions. We have updated the experimental methods to include these missing methods (pg.13) and have also included the imaging conditions the figure legend for Figure 1B (pg.10).

5. A broadened discussion of how the findings fit (or not) into the current model of convergent extension and roles of PCP in this process is needed. Is there any known connection between the PCP machinery and Cdh3 cis-clusters? How can PCP contribute to this mechanical heterogeneity along the mediolateral axis? Additionally, further discussion of where the directionality for contraction derives from is needed. Do clustered cadherins impose a directionality to the active vertex movements?

We have substantially revised the Discussion to include a more thorough treatment of these issues.

Reviewer #1:In follow-up to previous studies from the Wallingford group, the focus here is on junction dynamics in the anterior and posterior junctions of actively intercalating prospective notochord cells. The advances in characterizing vertex dynamics emerge from incorporating concepts from soft matter physics and microrheology. The conceptual framework borrowed from physics is an interesting move in the analysis of cell boundary motions in this classical system of directed cell rearrangement but I am not fully convinced that the framework brings new understanding to this problem. The joining of the conceptual framework of vertex analysis and the bicellular study of cis-cadherin would seek to bend principles borrowed from apical junction dynamics and apply to mesenchymal junction dynamics but it has not been made sufficiently clear what new insights this brings.1. What exactly is "mechanical heterogeneity" and what precise role does it play in cell intercalation and convergent extension? Is it necessary, sufficient, or merely coincident with the process? Answering these questions is critical to the potential impact of this paper on the field. There are many heterogeneous events and processes throughout a single embryonic event but why is this one so important?

Mechanical heterogeneity in this context refers to the differences in mechanical properties along a single cell-cell junction. It is true that Vanderleest et al. (2018, *eLife*), reported asymmetry and physical independence in the movement of two vertices bounding a single cell-cell junction. We nonetheless feel our manuscript is the first to directly demonstrate mechanical heterogeneity along the junction linking these two vertices.

Indeed, this novelty is significant enough that it has drawn the attention of the Gardel lab, leaders in the field. They have now posted a preprint in the BioRxiv reporting their own novel theoretical model to address the very mechanical heterogeneity along junctions that we describe here (https://www.biorxiv.org/content/10.1101/2021.02.26.433093v1). Critically, the work in that preprint uses optogenetics and cell culture, so provides an interesting complement to the in vivo work we present here.

Finally, and perhaps most crucially, we not only show that this heterogeneity exists in a developing vertebrate animal in vivo, but also that this heterogeneity is necessary for convergent extension and requires the function Cdh3 and PCP signaling.

We have expanded the Intro and Discussion of the paper to better emphasize this important point.

2. Very much like the observations from Vanderleest et al. (2018, eLife), the authors propose that motions of two vertices connecting shrinking junctions are independent; one vertex moves while the other remains fixed in place. This seems plausible but I have two issues with their analysis. The first is that the authors claim to use triangulation to track junctions in the "lab frame of reference", but I understand that triangulation requires 2 landmarks to track the absolute position of 1 point. The diagram in the supplement shows that 1 landmark is used to measure the position of the two ends of the v-junction; a second independent point is needed for true triangulation. The second issue is that the relative positions are not measured against a fixed spatial frame but rather against local structures whose movements may be coupled to movement of the tricellular junction of interest. Motions of yolk platelets or other tricellular junctions from one of the 3 cells surrounding a specific tricellular junction are likely to be mechanically coupled to that same junction. It would be prudent to use a platelet or junction from cells that are not one of these 3 cells.

The reviewer is correct about use of the word triangulation, and we have removed the term from the manuscript. We did however measure the MSD from three principally independent frames of reference. The first is a fixed coordinate system (lab frame of reference) where we used the pixels of the image to establish the coordinates with the upper left pixel set at (x,y)(0,0). The fixed lab frame of reference was used for the data in figure1 and Figure 1—figure supplement 2. In addition, we used two additional non-fixed landmarks to measure the MSD. Here we used yolk particles within the intercalating cell and neighboring vertices as the reference frame. As we observe similar MSD from each of these coordinate systems, one fixed and two internal landmarks, we confident in reporting our MSD results. These points have been clarified in the revision.

3. This reviewer enjoyed the descriptive focus on vertex kinematics from the microrheological perspective, however, some background on the pro's and con's of the methodology (e.g. Crocker and Hoffman, Meth Cell Bio, 2007) would be helpful. Furthermore, the level of processing to quantify vertex fluctuations passed over several key features of vertex motion that would be of great interest to the community and should be included. For instance, it would be very helpful to describe advective kinematics of these structures starting with displacement vs. time kymographs, and extending to velocity and persistence. Are there any correlations between the active vertex and lateral (or medial) vertex motions? I understand that these vertices move in a non-uniform manner but would appreciate more intuitive description of their motion as if they were, for instance, a set of cell protrusions. A detailed "protrusion-like" analysis has not been carried out for these structures and would be useful to compare to lamellipodial kinematics and leading edge fluctuations described at the substrate level of intercalating cells.

We agree this is an important issue. We note that persistence of motion is, in fact addressed in Figure 3E, F and that our newest efforts to understand protrusive activity are reported in our new preprint from Went et al.

4. This analysis suggests a novel function for tricellular junctions in mesenchymal tissues. In contrast to the tricellular junction of an epithelial sheet, this structure is not point-like but would need to be considered as a tricellular boundary that extends from the superficial, extracellular matrix facing cell surface to the most dorsal face of the notochord cell. Rather than a point, this junction is more like a knife-edge and can exhibit a complex topology typical of "escutoid cells" described by the Escudero lab (Nat Comms, 2018) or the Toyama lab (Nat Cell Bio, 2017). This raises questions about the position of the vertices measured and whether the kinematics are heterogeneous along all points of the junction, are they consistently active, or are some points advancing while others are retracting. Have the authors described kinematics of the same vertex at different z-positions?

We fully agree that an analysis of cell-cell junction mechanics along the length of the z-axes would be very informative, but we are limited both experimentally and within the scope of this single manuscript. The mesenchymal bi-cellular junctions and tricellular junctions do change depending on z-position, as is likely true with epithelial tissues, but we are limited by the time in which our microscopes can acquire t and z-stacks. The analysis in this manuscript required high temporal resolution which unfortunately prevented the acquisition of multiple z-positions. Also, while we take a bit of a reductionist approach looking at a single z-plane in this manuscript, we believe that this simplification of the biological problem allowed the detailed analysis of vertex movements which comprises the first half of the paper. An analysis of vertex movements at different z-positions will be an exciting future direction.

5. Several aspects of the methodology are missing. For instance, what criteria are used to distinguish shortening from non-shortening junctions? There are few details about the imaging conditions, such as the z depth of the images and the time interval between frames.

We apologize to the reviewer as some of the methodology was either missing or incorrectly ordered in the theory supplemental information (SI). We have added the missing methodology and reorganized the Appendix to correctly match the main manuscript

Specifically:

1. Criteria for distinguishing shortening vs non-shortening junctions. Any junction not shortening over a 400s timescale was considered non-shortening. We have added this criterion for junction shortening to the main text (pg.5) and to the Appendix (section 1).

2. Missing details of imaging conditions. We have updated the experimental methods to include these missing methods (pg.13) and have also included the imaging conditions the figure legend for Figure 1B (pg.10).

“Frames were acquired at a z-depth of 5μm above the ECM/coverslip and with a time interval of 2 seconds”

6. The authors stress that their model is unique in driving cell intercalation but there are several theoretical and computational models have previously demonstrated the role of either mediolateral cell protrusions or anisotropic contractile bicellular junctions in directing mediolateral cell intercalation. Multiple examples of these models including recent one by Belmonte and co-workers (PLoS Comp Bio, 2016) involve heterogeneous vertices, while others not requiring heterogeneous junctions, are sufficient in driving convergent extension. The compressed exponential fits the spatial and temporal scaled kinematic changes of the shrinking bicellular junction but it is not clear whether this is simply a mechanistically-based phenomenological principle or merely a "good-fit" to the data.

There are indeed many models of CE and we feel that all have value. However, the paper from Belmonte seeks to model purely filipodia-driven cell crawling. Work from the Sutherland and Toyama Labs, as well as our recent work in BioRxiv from Weng et al., show that CE in diverse animals is driven by an integration of cell crawling and junction contraction. It is thus that we specifically designed a model of vertex motion that was agnostic concerning the cell biological basis, and instead modeled simply the local mechanical environment. Just the same, we think this point deserve attention, and we have now included a discussion of the Belmonte model in the revision.

7. An alternative interpretation for presence of an active vertex is that it reflects the leading edge of a cell protruding mediolaterally between adjacent cells. Why is this possibility not discussed?

The interpretation that the active vertex is the protruding leading edge of a mediolateral cell was actually our first hypothesis prior to quantitatively analyzing the data. In fact, we collect a large dataset of time-lapse videos of actin at shortening v-junctions but when we analyzed this data, we found no correlation between actin accumulation, active/passive vertices, and junction shortening. We have included a new supplemental figure (2E-G) showing that actin at the active or passive vertex does not correlate with junction shortening. This issue is also addressed in more detail in the BioRxiv preprint by Weng et al.

8. The second half of the manuscript describes an important role for cis-cdh3 interactions in establishing or maintaining cell behaviors and polarity within converging and extending cells. The phenotypes are clearly demonstrated at the embryonic and tissue level. The subcellular phenotypes are also well described but it is not clear that the kinematic phenotypes are directly responsible for the failure of these cells to undergo directed cell rearrangement. Additionally, the cell autonomy of the defect is not clear. The authors would need to show that cis-interactions in anterior and posteriorly apposed cells alone are responsible for defects in rearrangement. For instance, would a wild-type lateral-cell, or one-lateral and one-AP cell be able to rescue polarity and active kinematics of the vertex?

As explained in more detail in our response to the essential revisions, above, we feel that a further exploration of Cdh3 function is beyond the scope of this already rather large paper.

9. Given the role of the actin cytoskeleton in directed cell intercalation I feel the role of cis-cdh3 interactions is only half-explored. What changes in protrusive activity, and actomyosin dynamics are perturbed? Are these specific to PCP polarity machinery operating in the mesoderm or can they also be observed in cis-cdh3 deficient non-polarized cells such as non-neural ectoderm cells (see Kim, J Cell Sci, 2011)? Up and down regulation of PCP signaling similarly regulates actomyosin contractility in both mesoderm cells and non-neural ectoderm – so this should be straightforward to test.

These are all outstanding questions, and we are excited to explore them. However, by endeavoring to make parallel contributions in both cell biology and physics, we have generated 50 panels of data in the main figures alone, and more than 30 additional panels of data in the supplement. Thus, straightforward or not, we feel that additional consideration of the cell biology here is beyond the scope of the current work.

10. Overall, more discussion is needed about how these kinematics would fit or not fit into the current model of convergent extension and roles of PCP in this process. Is there any known connection between the PCP machinery and Cdh3 cis-clusters? How can PCP contribute to this mechanical heterogeneity along the mediolateral axis?

We have now provided additional treatment of these points in the Discussion section of the manuscript.

Reviewer #2:[…] There are several potentially nice findings from this study. One weakness is that the methods (especially the computational methods) are not well-described, and the findings are not detailed well in the manuscript. It is often stated that a finding supports a particular hypothesis, followed by a reference to a figure, but there is not a detailed description of what the findings actually are and how this specifically supports the hypothesis. At times, this makes the manuscript read superficially. I believe the Results section needs substantial re-writing to more adequately explain the computational sections (~first half of manuscript). The Discussion is also brief in content. More specific comments follow below:1. I thought the authors had a bit of an odd way of introducing the work, especially the paragraph suggesting "a similarly granular understanding of subcellular mechanical properties" has not been achieved. (From Intro, "For example, the localization and turnover of actomyosin and cadherin adhesion proteins have been extensively quantified during *Drosophila* CE (Blankenship et al., 2006; Fernandez-Gonzalez et al., 2009; Levayer and Lecuit, 2013; Rauzi et al., 2008)), as have similar patterns for the Planar Cell Polarity (PCP) proteins and actomyosin during vertebrate CE (Butler and Wallingford, 2018; Kim and Davidson, 2011; Shindo and Wallingford, 2014). However, the significance of these molecular patterns remains unclear because we lack a similarly granular understanding of subcellular mechanical properties and their dynamics, which ultimately explain the cell behaviors that drive CE.".I think the authors have a potentially interesting study, but it does not appear to address a significantly different scale than what many studies out of the Lecuit lab (or others) have addressed (e.g., Fernandez-Gonzalez, Gardel, Campos, Dahmann, Kiehart, or Hutson labs).

We respectfully dispute this assertion. While from the standpoint of basic physics the scales are similar, there are several biological distinctions that make our work both novel and highly significant:

1. Of the six labs mentioned above, five work exclusively with *Drosophila* and the sixth with cultured epithelial cells. A broader exploration of the issue in other settings, such as the vertebrate animal *Xenopus*, is critical so that we may learn which elements of *Drosophila* CE are universal and which, like the fly itself, are evolutionarily derived.

2. Moreover, all of the work referenced above deals only with well-behaved polarized *epithelial* cells, while our work is the messier context of *mesenchymal* cells. We think our extension from one tissue type to another is cell biologically very important. This is especially significant, because we *did not* expect the mesenchymal cells that we studied to share so many features with previously studied epithelial cells!

3. Perhaps most critically, neither *Drosophila* nor cultured cells rely on PCP signaling for CE, yet PCP signaling is essential for both mesenchymal and epithelial CE in all vertebrate embryos, as well as in several vertebrate organ systems. Thus, PCP-mediated CE is an essential biological process that cannot be studied in *Drosophila* or in cultured cells, making our new findings in *Xenopus* highly significant.

4. Finally, human birth defects are the most lethal diseases of children in the US (twice as lethal as pediatric cancers) and PCP genes are among the most well-defined genetic risk factors for human neural tube defects. Thus, understanding the biomechanics and cell biology specifically of PCP-dependent CE, which again cannot be explored in flies of cell culture, is highly significant.

It would be appropriate to better acknowledge prior work – it does not detract from the accomplishments of this study. My apologies, but at times the writing, especially the physical overlay of the paper, sounds dismissive of other works.

We are very sensitive to this concern. We have now added substantially more verbiage and several additional citations, both in the Introduction and in the Discussion, in an effort to address it.

2. It is not always clear how some of the biophysical analysis are performed and/or support the authors major contentions. One example, "We found that v-junction shortening was dominated by the movement of a single "active" vertex, while the other "passive" vertex moved comparatively less (Figure 1C, D)(SI, Section 1). Three distinct metrics demonstrated that this asymmetry was not a point-of-reference artifact (Supp. Figure 2)."While I suspect they are correct, this is one major sticking point and it appears the authors are not meeting scientific standard in demonstrating this point in its current form. First, the wording 'Three distinct metrics demonstrated that this asymmetry was not a point-of-reference artifact' seems to suggest that the authors are showing three independent pieces of evidence, but this is not the case (maybe this impression was not intended, so maybe reword?). What the authors are actually showing are the MSD results in three different coordinate systems (the 'lab' frame, the 'cell' frame which is a reference frame relative to a yolk particle, and the reference frame relative to a non-moving neighboring vertex). As best as I can guess, this is the comparison of the results from the three reference frames that would (potentially) allow the conclusion that the different motion in the two vertices is not an artifact, but this is not possible to tell, because the authors only describe their results in the supplementary figure, and do not explain anywhere how/why these three different results prove the point that the different motion cannot be a reference point artifact. The mere fact that the authors were able to identify three different reference frames in which the vertices move differently does not meet this threshold; they would have to explain better why one of them (e.g. the cell frame) represents something like a fixed local coordinate system, and – for example – show that an ensemble of yolk particles doesn't 'deform' significantly over the associated time scales (for example, they could show that the yolk particles only minimally changes their interparticle distances during this time?).Additionally, the authors write in the figure legend 'We refer to this as the relative MSD in the "cell" frame of reference, which eliminates the effect of translational and rotational motion of the tissue during time-lapse imaging.' What the authors show in the figure is a single yolk particle as reference point, and a single reference point would only be able to eliminate the effect of translation, but not translation and rotation at the same time. So are the authors using only a single yolk particle for a set of two vertices and ignoring rotation effects? Or are they using a system where they can construct a coordinate system from multiple yolk particles simultaneously (which could correct for both translation and rotation) but not showing this procedure in the figure? And going forward, which reference frame are the authors using for their analyses, since all three show different results, and why? These things would be best explained explicitly in a section of the supplement.

These are crucial points. As discussed in detail in response to a similar comment, above, this concern stems from a lack of clarity in our initial submission. A new schematic in Figure 1—figure supplement 2 clarifies that we used a fixed frame of reference and a coordinate system for the key measurements in Figure 1. We also included the MSD for the yolk particle and the neighboring vertex Figure 1—figure supplement 2, compared to the fixed reference frame, to show minimal movement of these structures compared to the vertices. Finally, we removed language concerning “triangulation” and “translation and rotation of motion” as they were incorrectly used. Instead, we have clarified that we measured the MSD from three principally independent frames of reference, one fixed and two that are local within the image.

3. Where does the directionality for contraction derive from? I found the authors "transverse" vs "in-line" coordinate system somewhat confusing. Transverse in other studies has often been applied to the non-contracting interface, but lines drawn in figure schematics seem to indicate a more cell-center axis. I believe the authors are showing more clustered, stable adhesions correlating with "active" vertices, but does this impose a directionality to the active vertex movements (in the experimental data sets)? Any further description would be helpful, and this could also be expanded on in the Discussion.

We have made an effort to address this complex issue in the revised discussion, though we note that full understanding is still a long way off!

Also, we do note that transverse in our context is different from that used in *Drosophila* where transverse junctions (t-junctions) are those perpendicular to the shortening vertical junctions (v-junctions). We have maintained the use of t,v-junctions to be consistent with work done in *Drosophila* but do appreciate that this may add to some confusion. Here we used transverse fluctuations to describe vertex motion that is perpendicular to the shortening junction. We have added text to main manuscript (pg.6) to help clarify our use of the term transverse. The theory appendix (section 12) also provides a more thorough quantitative description of transverse fluctuations which we hope clarifies this point.

4. Some of the MSD difference between active and passive vertices appear small. It is not clear how we know that the differences between active and passive vertices represent something 'real' and not an artifact of the classification process. In the first section of the results, the authors are comparing various metrics (such as MSD) between 'active' and 'passive' vertices, but I am not finding a single clear explanation of how these categories were defined. Are the authors using a threshold to classify a given vertex as active or passive? Or are the authors taking vertex pairs, and splitting up each pair into one active and one passive partner, based on MSD magnitude? This is a critical piece of information for interpreting the results. Can the authors show that there are two 'modes' of behavior (active and passive) using a clustering approach of some type?

The classification method for the active and passive vertex is defined in the theory appendix section 1. As noted above, our initial submission had some organizational errors the theory appendix which have been fixed for the resubmission. We apologize and ask reviewers to please see the updated theory appendix. We also appreciate that defining the active and passive vertex is a key element of this manuscript and have updated the main manuscript with this definition (pg.4).

5. I would like to hear more interpretation of what the mechanical properties of the cells are that are being affected in the -cis binding mutants (this is taken somewhat as canon, but does it behave the same in these *Xenopus* explants?). I know the authors say cell-cell adhesion is (somewhat?) rescued by this -cis mutant and embryos do not fully dissociate. Can the authors better detail the degree to which the strength of cell-cell adhesion is rescued versus local properties like the asymmetry of adhesive forces at cell vertices?

We feel that these questions, while important, are beyond the scope of the already large paper here.

6. A final note about the modeling. About their model predictions, the authors state that 'Our theory makes a curiously counter-intuitive prediction: that the more fluid-like motion of the active vertex occurs in the context of increased local stiffness (i.e. higher viscoelastic parameter), while the more glass-like motion of the passive vertex occurs in a relatively decreased stiffness regime.'I am not sure that there is something counter-intuitive about this finding. While it is true that in the classic model for epithelial morphology (Farhadifar et al. 2007), the more fluid-like behavior is associated with lower stiffness, that's because it's related to adhesion forces dominating over elastic forces, so the asymmetric vertex movement is being driven by the minimization of adhesion energies. In the authors' model, there are no adhesion forces, and the vertex displacement is driven by the external piston forces – so it seems logical that the more effective motion (in the active vertex) would occur in the case where the higher spring stiffness allows better force transmission (and the passive vertex is the one where the forces dissipates more quickly in the softer environment). Thus, the fluidization arguments from other models/systems in the literature would not seem to fully apply to this system.

We understand this sentiment, but after several presentations, we are acutely aware that this result is in fact quite counter-intuitive to cell biology audiences! Just the same, we have removed the term counter-intuitive from the main manuscript.

7. "Symmetrical clustering in non-shortening reflects the symmetrical dynamics of vertices bounding these junctions" – I believe "reflects" suggests a direct connection, whereas this is more of a "correlates" or "suggestive" statement.8. Figures often have panel titles like "Deviation from compressed exponential" – it would helpful to better title these to indicate the rough property that the authors are trying to report on, and put more specificity into the figure legends.

We appreciate this comment and have spent a significant amount of time trying to create useful titles that satisfy both the physicists and biologists. We settled on titles that use the same language as the theory appendix where we define most of the graphs. Our hope is that the theory appendix can clarify any questions where the content of the graph is not entirely intuitive.

Reviewer #3:[…] I suspect that this study is not the end of the matter. In particular, it would be important to understand how local differences in cadherins affect junctional mechanics. The authors cite earlier work from the Mege lab indicating that cadherins can influence membrane mechanics. Whether this is at the level of the bilayer or via alterations in the cortical cytoskeleton would be important to address. But this is beyond the reasonable scope of a paper; indeed, it would speak to the potential for this study to motivate future research.1. Figure 4G. Is this correct as currently presented? The authors describe this as showing that cadherin cluster size near non-shortening vertices is the same as that around the passive vertices in shortening junctions. But to my eye, what is drawn as the non-shortening data set looks similar to the set that is labelled as (active). Has something been mislabelled?

The data in Figure 4G are not incorrectly labeled. The difference between the active and non-shortening junctions is best observed by focusing on the upper end of the distribution where the active violin plot is noticeably wider than the non-shortening violin plot.

2. Embryo cohesion. This is measured as embryo survival at stage 23. But that seems to me to be a very indirect measure of tissue integrity. Changes in survival may reflect catastrophic loss of integrity, but miss more subtle features. I would be happy enough if they were to use the term "viability" instead of "cohesion". There is still a very interesting different in effect of the cis-mutant.

We appreciate this comment and have struggled with finding the correct term ourselves. Initially we used the term embryo survival instead of embryo cohesion but found this to but somewhat misleading. While the cis-mutant embryos do survive well past the chd3 knockdown embryos they do not survive to make adult animals. The updated manuscript now uses the term embryo integrity in place of embryo cohesion as integrity has less of a direct connection to adhesion than the term embryo cohesion.

3. I think that some of the analyses could be explained more intuitively for the general reader. For example, when the authors introduce their Straightness Index, it is not clear from the Results that they are talking about the straightness of motion of the vertices. (For junction folks, "straightness" sounds like the measures of junctional straightness that have been used as proxies of line tension.) It would only take a sentence to give an intuitive explanation in this and other instances.

As noted above we have updated the manuscript at a number of points to try and clarify the language for a general reader, including the section discussing the straightness index. For example:

Definition of Straightness Index. The following text was added to the main manuscript (pg.6) to clarify that Straightness Index referred to vertex motion and not bicellular junction “straightness”.

“Analysis of the straightness index, quantifying how straight vertices move along the in-line direction”.

The complete definition of Straightness index is given in the Appendix (section 12).

4. Glass-like behaviour in junctions. To what extent can one really say that the passive junctions are "more glass-like". If I read Figure 1E correctly, the exponent of "t" for the passive junctions is still >1 (1.3 from Figure 1E). So, is it more helpful to simply regard them as less fluidized, rather than invoking subdiffusive behaviour (which would require the exponent to be <1)? I'm happy to the corrected by the authors (or the other reviewers), but beyond a point of phenomenology I'm still not sure that I can see an underlying intuition that would be really helpful.Along these lines, is it possible to statistically compare the differences in the exponentials?5. "Non-shortening" junctions. Were these tested away from junctions that shortened? What about the junctions that also contribute to the vertices at the ends of the junctions that shortened? Did these show any asymmetry in mechanics or in cadherin clustering?

A good question. The non-shortening junctions were tested away from the junctions that shortened and we have not performed a complete analysis comparing “close” versus “far” non-shortening junctions. We will consider this in future work.